# Time-R1: Post-Training Large Vision Language Model for Temporal Video Grounding

Ye Wang[1*]   Ziheng Wang[1*]   Boshen Xu[1*‡]   Yang Du[1]   Kejun Lin[1]   Zihan Xiao[1]
Zihao Yue[1]   Jianzhong Ju[2]   Liang Zhang[1]   Dingyi Yang[1]   Xiangnan Fang[1]   Zewen He[2]
Zhenbo Luo[2]   Wenxuan Wang[1]   Junqi Lin[2]   Jian Luan[2]   Qin Jin[1†]

[1]AIM3 Lab, Renmin University of China    [2]MiLM Plus, Xiaomi Inc
Project Page: https://xuboshen.github.io/Time-R1/

## Abstract

Temporal Video Grounding (TVG), the task of locating specific video segments based on language queries, is a core challenge in long-form video understanding. While recent Large Vision-Language Models (LVLMs) have shown early promise in tackling TVG through supervised fine-tuning (SFT), their ability to generalize remains limited. To address this, we propose a novel post-training framework that enhances the generalization capabilities of LVLMs via reinforcement learning (RL). Specifically, our contributions span three key directions: (1) **Time-R1**: we introduce a reasoning-guided post-training framework via RL with verifiable reward to enhance capabilities of LVLMs on the TVG task. (2) **TimeRFT**: we explore post-training strategies on our curated RL-friendly dataset, which trains the model to progressively comprehend more difficult samples, leading to better generalization. (3) **TVGBench**: we carefully construct a small but comprehensive and balanced benchmark suitable for LVLM evaluation, which is sourced from available public benchmarks. Extensive experiments demonstrate that Time-R1 achieves state-of-the-art performance across multiple downstream datasets using significantly less training data than prior LVLM approaches, while improving its general video understanding capabilities.

## 1 Introduction

Understanding long-form videos has long been a core ambition in computer vision [14, 25, 9]. A critical step toward this goal is Temporal Video Grounding (TVG) [15, 66], which enables models to localize video segments corresponding to natural language queries (e.g., "Find the segment where a person walks into the living room."). Such capability is fundamental for real-world applications, including smart home assistants [60, 16, 49] and video retrieval systems on online platforms [5, 3].

Traditional TVG approaches adopt a feature-based paradigm, where pretrained models (e.g., CLIP [45], I3D [6]) extract text and video features, followed by task-specific grounding models [31, 23, 30]. However, these methods suffer from error accumulation due to imperfect pretrained features. To overcome these limitations, recent efforts have shifted toward end-to-end Large Vision-Language Models (LVLMs) [47, 65], which directly process long-form videos and text queries. Despite being pretrained on datasets 100× larger than domain-specific benchmarks [49], LVLMs (with 7B+ parameters) often underperform compared to much smaller feature-based models (e.g., 9M parameters [23]). This raises a critical question: Why do LVLMs, despite their vast pretrained knowledge, fail to excel on TVG?

---

† Corresponding author: Qin Jin; * Equal contribution, listed in alphabetical order; ‡ Project lead.

39th Conference on Neural Information Processing Systems (NeurIPS 2025).

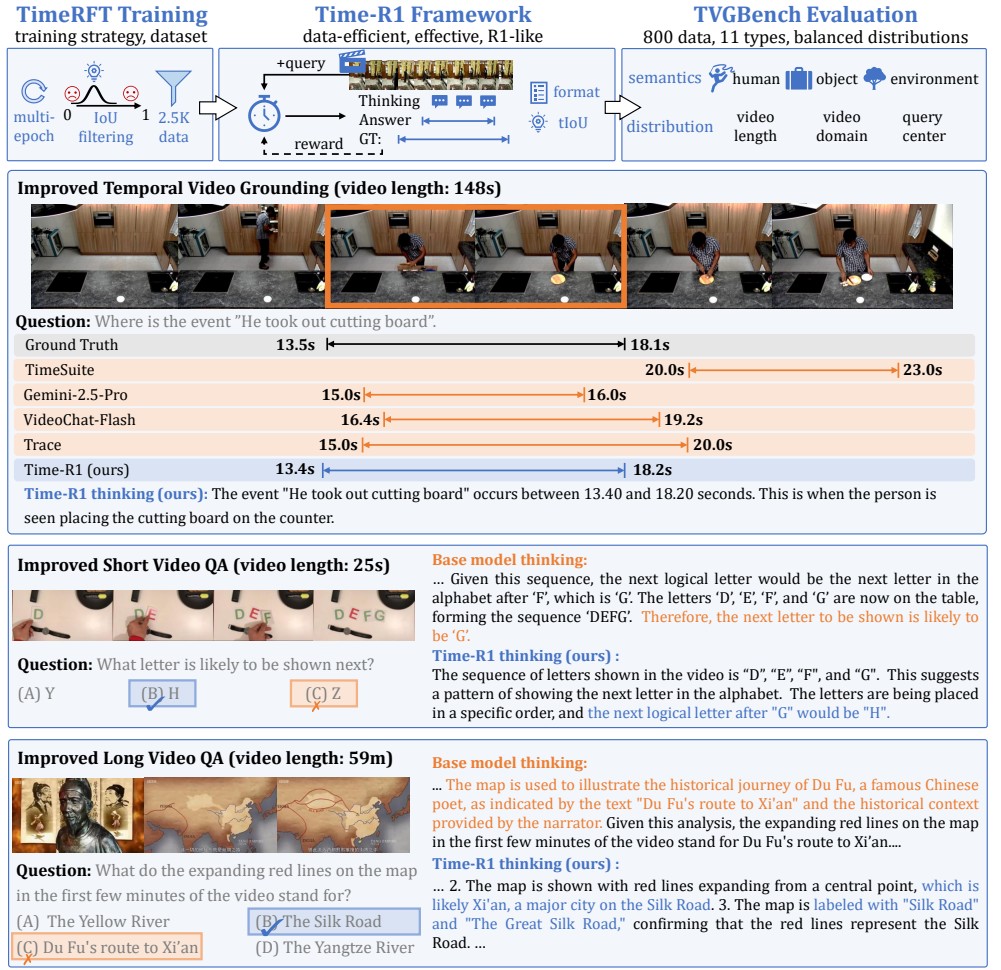

Figure 1: Our contributions include a novel post-training framework for LVLMs via reinforcement learning, **Time-R1**; an RL fine-tuning strategy along with its associated training dataset, **TimeRFT**; and a new benchmark, **TVGBench**, for evaluating LVLMs on the TVG task. Our Time-R1 model not only achieves SoTA TVG performance, but also enhances performance on both short- and long-form multi-choice video question answering tasks.

We attribute the suboptimal performance of LVLMs to over-penalization of false negatives during supervised fine-tuning (SFT). For instance, when the ground truth timestamp is [2s, 4s], even when the model makes a reasonable prediction of timestamp [1.9s, 3.9s], the autoregressive loss would still be undesirably high. Such disproportionate penalties on reasonable predictions result in overfitting and poor generalization. While previous solutions have attempted to address this by introducing new timestamp tokens into the vocabulary [19, 17, 58] or by appending a regression head to predict timestamps [70], they often sacrifice the pretrained language capabilities of LLMs.

Inspired by recent success in reinforcement learning (RL) for post-training LLMs [41, 1] with chain-of-thought (CoT) prompting, especially in domains with deterministic answers, such as code generation and mathematical reasoning, we explore whether RL can serve as an effective alternative for TVG. Unlike SFT, RL allows direct optimization of task-specific metrics (e.g., IoU), thereby reducing rigid penalties of autoregressive losses and encouraging plausible timestamp predictions. In this work, we present an RL-based framework Time-R1 that effectively post-trains LVLMs for TVG and pushes the performance frontier. Our contributions include:

• **RL-based framework for temporal video grounding.** We introduce **Time-R1**, a reasoning-enhanced post-training framework via RL with verifiable rewards, where the LVLM first generates chain-of-thought descriptions and then predicts timestamps. The post-training process is optimized

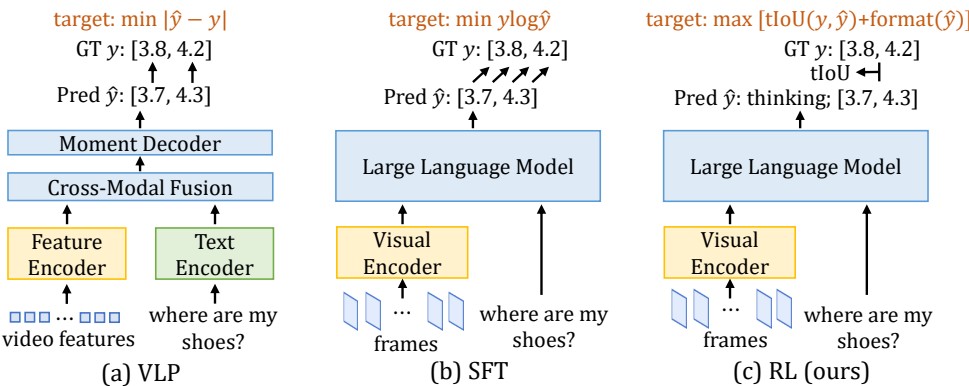

Figure 2: Comparison of different approaches for the TVG task, including feature-based video-language pretraining (VLP) [39, 23], supervised fine-tuning (SFT) [44, 65], and RL (ours).

using Generalized Reinforcement Policy Optimization (GRPO) with a novel reward function, incorporating both a structured template reward and a timestamp-aware tIoU reward.

• **Time-aware reinforcement fine-tuning.** We propose **TimeRFT**, a reinforcement fine-tuning strategy with dynamic hard sampling, which mines hard samples on a curated dataset and progressively selects low-IoU samples for multi-epoch training. To ensure stable reasoning and reduce hallucinations, we adopt a cold-start approach to generate CoT with video captions. To support RL-friendly training, we curate an RFT dataset with difficulty annotations on the TVG task.

• **Comprehensive benchmark for LVLMs on TVG.** Existing TVG benchmarks are designed for the large-scale evaluation of small models. Considering the inference speed bottlenecks and general-purpose role of LVLMs, we construct **TVGBench**, a compact yet comprehensive benchmark for TVG. We carefully balance the video distribution, query distribution, and design specific query semantics to ensure that the benchmark is well-suited for evaluating LVLMs.

• **State-of-the-Art results and generalization.** Compared with 7B LVLMs on the temporal video grounding task, our method outperforms all prior SFT-based methods. After fine-tuning on downstream benchmarks, it surpasses many previous feature-based approaches. Furthermore, Time-R1 also improves the model's general video understanding on video QA benchmarks.

## 2 Related Works

**Temporal video grounding.** The TVG task [15, 3] aims to localize temporal segments in untrimmed long videos given natural language queries. Previous works can be broadly categorized into feature-based video-language pretraining and frame-based LVLM methods, as shown in Figure 2. Feature-based methods first extract video and language features using pre-trained encoders (e.g., I3D [6], EgoVLP [30], CLIP [45], BERT [12] etc.), then build timestamp prediction modules based on multimodal fused features. Existing methods differ mainly in their design of the multimodal fusion module and timestamp prediction module. For example, SnAG [39] adopts a late fusion strategy and regresses timestamps directly in a single forward pass without proposal generation. While effective, these approaches are fundamentally limited by the quality of the pretrained features. Recent efforts have shifted toward end-to-end frame-based methods by fine-tuning LVLMs using SFT with autoregressive losses [28, 47, 65, 20, 55, 19, 26]. For instance, TRACE [19] treats each event as a combination of timestamp, saliency score, and caption, and fine-tunes the LVLM to generate event sequences autoregressively. However, such methods still underperform on even simple TVG benchmarks like Charades [49], often lagging behind feature-based approaches. In this work, we propose a novel RL-based post-training framework that establishes new state-of-the-art performance for LVLMs on TVG tasks, even surpassing many feature-based methods.

**RL in LLMs and LVLMs.** RL is a foundational machine learning paradigm applied in diverse domains such as game playing [50], robotics [36], and increasingly, language and vision-language models [41]. RL updates models by interacting with environments and maximizing reward signals. In recent years, RL has profoundly affected the field of LLM and LVLM post-training which falls into two main categories: Reinforcement Learning with Human Feedback (RLHF) [42, 62] and Reinforcement Learning with Verifiable Reward (RLVR) [1, 41, 7]. Early works find that RLHF

can align LLM to generate human preference data, which primarily reduces the safety risks in LLM and hallucination problems in LVLM. For example, RLHF-V [62] collects fine-grained pairs of incorrect and corrected captions and leverages direct preference optimization to optimize the model to generate correct captions, thus mitigating hallucinations. More recent works have explored RLVR in tasks with deterministic answers, which not only benefits mathematical problem solving and code generations in LLMs (e.g. DeepSeek-R1 [1]), but also enhances the generalization of LVLMs across a range of applications, such as visual grounding [34] and visual reasoning [51]. As a pioneer among open-source LLMs, DeepSeek-R1 [1] adopts GRPO to enhance reasoning capabilities by designing rule-based rewards that incorporate both reasoning templates and final answers. In the context of LVLMs, MM-Eureka [37] investigates multimodal image reasoning using GRPO, introducing an online filtering mechanism and a two-stage training strategy to stabilize the optimization process. However, existing approaches remain confined to language [1, 61], image understanding [7, 51, 34, 37], and short video understanding [69, 29]. It remains unclear whether and how reinforcement learning impacts long-form video understanding. To bridge this gap, we take a first step by introducing RLVR into LVLMs for the temporal video grounding task.

## 3 Method

The TVG task aims to temporally localize video segments within long-form videos based on natural language queries. Given a video of duration $t$ seconds, which is represented as a sequence of $T$ frames $\{x_1, \ldots, x_T\}$, and a language query $q$, the goal is to identify the temporal boundaries $[t_s, t_e]$ of the segment that best corresponds to $q$, where $t_s, t_e \in \mathbb{R}^+$. In this work, we introduce Time-R1, a framework designed to unleash the potential of LVLMs for the TVG task using RL. We first provide background on RL-based training for LLMs in Section 3.1, then detail the training procedure of Time-R1 in Section 3.2. Next, we describe specific training techniques used in TimeRFT in Section 3.3, and finally, we present the construction of our evaluation benchmark TVGBench in Section 3.4.

### 3.1 Background of GRPO: RL for LLM

As a pioneer among open-sourced R1-style LLMs, Deepseek-R1 [1] leverages GRPO to train the policy model $\pi_\theta$ (i.e., the LLM) to think before answering, making it particularly well-suited for tasks with well-defined answers, such as mathematical reasoning. In the GRPO framework, given an input question $p$, the LLM samples $G$ candidate responses $o = \{o_1, \ldots, o_G\}$, and a reward function $r(\cdot)$ assigns a reward score to each response, yielding $\{r(o_1), \ldots, r(o_G)\}$. GRPO encourages the LLM to generate responses that maximize a weighted sum reward $R(o)$, defined by:

$$R(o) = \sum_{i=1}^{G} \frac{\pi_\theta(o_i)}{\pi_{\theta_{\text{old}}}(o_i)} \cdot \frac{r(o_i) - \text{mean}(\{r(o_i)\}_{i=1}^{G})}{\text{std}(\{r(o_i)\}_{i=1}^{G})} \tag{1}$$

where $\pi_\theta(o)$ denotes the probability of LLM generating the response $o$, and $\pi_{\theta_{\text{old}}}$ represents the LLM parameters from a recently optimized state. To ensure training stability and avoid large deviations from the original language model behavior, the final training objective incorporates a KL-divergence regularization term [1], penalizing divergence between $\pi_\theta$ and $\pi_{\text{ref}}$:

$$\max_{\pi_\theta} \mathbb{E}_{o \sim \pi_{\theta_{\text{old}}}(p)}[R(o) - \beta \text{D}_{\text{KL}}(\pi_\theta \| \pi_{\text{ref}})] \tag{2}$$

where $\beta$ is a scaling coefficient that balances reward maximization and policy stability. We omit the clipping operation for simplicity.

### 3.2 Time-R1: RL for Temporal Video Grounding

Since the TVG task has defined answers and well-established evaluation metrics, RL can optimize LVLMs for task-specific performance through tailored reward design. To enhance interpretability and align with human-like reasoning, we additionally incorporate an explicit "thinking process" prior to timestamp prediction. This process encourages the model to produce contextualized video descriptions that support its final decision. We detail our reward modeling and training process below.

**Reward modeling.** The reward $r_i$ plays a crucial role in guiding the model's learning objective. To encourage effective temporal grounding with an explicit reasoning process, we design a composite

reward function comprising two components: the timestamp-aware Intersection over Union (IoU) $r_{\text{tIoU}}$ and the reasoning template reward $r_{\text{form}}$.

• **Timestamp-aware IoU reward** $r_{\text{tIoU}}(\cdot)$. The TVG task primarily uses IoU [63] to evaluate the quality of predicted segments against the ground-truth $[t'_s, t'_e]$, computed as:

$$\text{IoU} = \frac{[t_s, t_e] \cap [t'_s, t'_e]}{[t_s, t_e] \cup [t'_s, t'_e]} \tag{3}$$

where $A \cap B$ and $A \cup B$ denote the union and intersection between sets A and B, respectively. Optimizing for the IoU inherently encourages the LVLM to produce predictions that fall within a permissible range of variation $\epsilon$, such that $t'_{s \text{ or } e} - \epsilon \leq t_{s \text{ or } e} \leq t'_{s \text{ or } e} + \epsilon$ still yields high IoUs. This encourages the LVLM to focus more on the semantic understanding of the event within possible temporal boundaries, rather than rigidly requiring exact temporal alignment like SFT. However, standard IoU may fail to accurately reflect the quality of temporal alignment in certain scenarios. For example, when the ground truth span is [0, 30] (i.e., the full video), any prediction covering more than 30% of the video would result in an IoU greater than 0.3. A prediction like [10, 25] would yield an IoU of 0.5, which overestimates its quality despite incorrect timestamps. To address this issue, we introduce the timestamp-aware IoU (tIoU) as a corrective measure. tIoU augments the standard IoU with penalties on timestamp deviations, defined as:

$$r_{\text{tIoU}}(o) = \text{IoU} \cdot (1 - \frac{|t_s - t'_s|}{t}) \cdot (1 - \frac{|t_e - t'_e|}{t}) \tag{4}$$

This modification penalizes predictions that deviate from the reference timestamps relative to the video duration $t$. In the earlier example, the reward value changes from 0.5 (IoU) to 0.28 (tIoU), providing a more realistic signal for learning. Overall, tIoU acts as a stricter and more informative reward signal, encouraging the LVLM to develop a deeper temporal understanding of events, rather than relying on superficial shortcuts.

• **Reasoning template reward** $r_{\text{form}}(\cdot)$. In TVG, the video segments relevant to a textual query typically comprise only a small portion of the entire long video. For LVLMs, it is therefore suboptimal to directly predict timestamps without first engaging in a reasoning process to identify the relevant content. Instead, the model should allocate its computational capacity toward reasoning over visual and linguistic cues to better understand the temporal context before making predictions. For instance, given the query "the man washes dishes", reasoning that the person is likely in a kitchen can improve temporal localization. Such context-aware inference supports more accurate and semantically aligned predictions. To encourage this behavior, we introduce a template-based reasoning reward, which incentivizes the model to generate intermediate reasoning steps (structured in a predefined format) prior to timestamp localization. The reasoning template reward requires the LVLM to structure its response like "<think>···</think> <answer><$t_s$ to $t_e$></answer>", formulated as:

$$r_{\text{form}}(o) = \begin{cases} 0, \text{if } o \text{ has wrong fromat} \\ 1, \text{if } o \text{ has correct fromat} \end{cases} \tag{5}$$

The overall reward is the sum of the two:

$$r(o) = r_{\text{tIoU}}(o) + r_{\text{form}}(o) \tag{6}$$

**GRPO training.** The LVLM $\mathcal{F}(\cdot)$ takes the video frames $x_1, \ldots, x_t$ and the language query $q$ as input and generates $G$ candidate responses $o_1, \ldots, o_G$, where each response is computed as $o_i = \mathcal{F}(x_1, \ldots, x_t; q)$. The reward for each response is calculated using Equation 1, and the model is optimized with the GRPO objective in Equation 2. To focus learning on the reasoning and localization capabilities, we freeze the visual encoder and update only the parameters of the LLM during training.

### 3.3 TimeRFT: Time-Aware RL-Friendly Fine-Tuning

Due to the high computational cost associated with RL training, we explore data-efficient strategies to reduce sample requirements. To this end, we propose TimeRFT, which incorporates time-aware, RL-friendly dataset curation and fine-tuning techniques aimed at enhancing generalization while minimizing training overhead.

**RL-friendly dataset curation.** We construct the TimeRFT dataset by leveraging only TVG samples, and assign a difficulty score to each sample based on the base model's performance. A small subset is then selected for subsequent RL training.

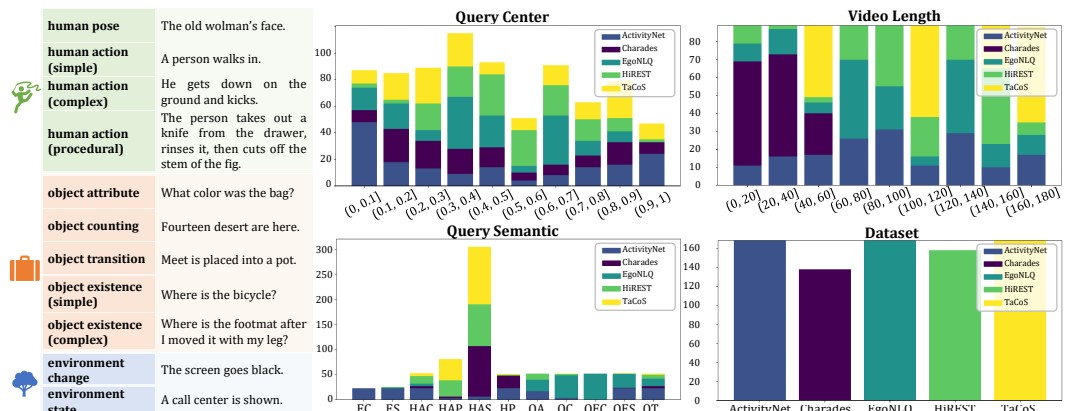

Figure 3: Statistics of TVGBench. TVGBench comprises 11 types of queries covering aspects related to humans, objects, and environments. As illustrated in the figure on the right, the distributions of query center, video length, and dataset source are designed to be as balanced as possible. This balanced construction allows for a comprehensive evaluation of model performance across different dimensions, enabling fine-grained analysis along each axis during benchmarking.

- **Source data collection.** Our training videos are sourced from Internet video datasets including YT-Temporal [59], DiDeMo [3], QuerYD [40], InternVid [52], and HowTo100M [38]. We obtain grounding data with annotations from VTG-IT [17], TimeIT [47], TimePro [65], HTStep [2], and LongVid [28]. This process yields 339K temporal grounding samples.
- **RFT data filtering.** We propose a data selection strategy based on training difficulty to significantly reduce training costs while preserving strong generalization performance. Models trained only on easy samples (e.g., IoU $\geq 0.7$) tend to overfit, whereas training on overly difficult samples (e.g., IoU $= 0$) often suffers from sparse reward signals, making it hard for the model to receive positive feedback. To strike a balance, we select samples of moderate difficulty that are more conducive to generalization during reinforcement fine-tuning. We first estimate a difficulty score for each sample based on the performance of the base model. For grounding tasks, difficulty is quantified using the IoU between the predicted and ground-truth temporal regions. We then filter out samples that are either too easy or too hard. Specifically, we sample a subset of data from a Gaussian distribution over the IoU axis centered at 0.3, resulting in a set of 2.5K moderately difficult samples for RL training.

**RFT training strategy.** For selected difficult samples, the model may struggle to learn them in a single pass. However, we argue that effectively mastering these challenging cases is essential for improving overall model performance. To this end, we employ a multi-epoch training approach combined with per-epoch sample filtering, allowing the model to repeatedly focus on harder samples and gradually improve its understanding.

- **Dynamic hard sampling.** We adopt a multi-epoch training strategy coupled with per-epoch sample filtering to enhance learning from difficult examples. The model is trained over multiple epochs, and after each epoch, we exclude easy samples with an IoU greater than 0.7 that have become easy. This dynamic curriculum discourages overfitting on easy instances while ensuring consistent exposure to harder samples, ultimately promoting stronger generalization.
- **Cold start fine-tuning with few CoT data.** For smaller models (e.g., 3B parameters), we observe that directly training with RL to generate CoT responses often results in reasoning steps that are either unintelligible or hallucinated, which impairs answer quality. Additionally, the length of generated reasoning during early training stages is difficult to control, leading to an unstable training process. To address these issues, we introduce a cold-start fine-tuning strategy using a small set of CoT-formatted examples that encourage grounded reasoning aligned with video content. Specifically, we guide the model to produce structured sequential captions with associated timestamps. The SFT template is defined as:

$$< \text{think} >< t_{s_1} \text{ to } t_{e_1} : C_1; \ t_{s_2} \text{ to } t_{e_2} : C_2 >< /\text{think} >< \text{answer} > t_s \text{ to } t_e < /\text{answer} > \quad (7)$$

where $C_i$ represent captions corresponding to video segments $[t_{s_i}, t_{e_i}]$, respectively.

### 3.4 TVGBench: Evaluation Benchmark for LVLM on Temporal Video Grounding

Existing benchmarks for temporal video grounding either focus on large-scale datasets tailored for smaller models within specific domains (e.g., human activities in ActivityNet) or consist of small, limited test sets (e.g., the 2K home activity samples in Charades) typically used for LVLM evaluation due to their slower inference speed. However, these benchmarks fall short in capturing the evaluation needs of LVLMs, which, despite slower inference, exhibit strong generalization capabilities. To bridge this gap, we introduce TVGBench, a lightweight yet comprehensive evaluation benchmark specifically designed for assessing the performance of LVLMs on temporal video grounding tasks.

**Data sources.** To ensure a comprehensive evaluation, we construct our TVGBench with curating samples from five public benchmarks with a balanced distribution of data source: Charades-STA [49], ActivityNet-Captions [5], HiREST [64], EgoNLQ [16], and TaCoS [46].

**Benchmark statistics.** We carefully balance the video duration, video domain, query center and construct query semantics in TVGBench to construct 800 instances, as shown in Figure 3.

• **Distribution statistics.** Video durations in the dataset have a balanced range from short clips up to 3 minutes long. To ensure temporal diversity, the center timestamps of queries are designed to be approximately uniformly distributed across the entire span of each video.
• **Query semantics.** Since the original datasets do not provide semantic labels for queries, we manually define 11 semantic categories grouped under three major types: human, object, and environment. We leverage DeepSeek-V3 [11] to annotate the semantic type of each query and ensure approximate balance across these categories. While most categories are evenly represented, the Human Action Simple (HAS) category is slightly overrepresented due to inherent dataset bias (simple indoor actions are more common). In such cases, we prioritize achieving a balance across datasets from different domains while maintaining semantic diversity, accepting a skew in HAS.

## 4 Experiments

We first present our experimental setup in Section 4.1. Then, we evaluate our model from three key perspectives: (1) Comparison with state-of-the-art methods in Section 4.2: We evaluate our model across multiple TVG benchmarks to assess its performance against existing approaches; (2) Ablation studies and analyses in Section 4.3: We examine the individual contributions of each component in our framework to better understand their roles in overall performance. We also compare RL and SFT strategies across TVG, short video QA, and long video QA tasks.

### 4.1 Experimental Setup

**Benchmarks.** We evaluate our model on a wide range of benchmarks covering both temporal video grounding and general video understanding tasks, including: (1) *Charades-STA* [49] contains 6,672 long videos capturing indoor human activities. The official split for the TVG task includes 12,408 clip-query pairs for training and 3,720 for testing. (2) *ActivityNet* [5] comprises 20K long videos with an average of 3.65 clip-query pairs per video. Following previous work in fine-tuning setting [67, 23] for the TVG task, we use the standard dataset splits with 37,421 training, 17,505 validation, and 17,031 test samples. (3) *MVBench* [27] is a short video QA benchmark focused on temporal reasoning. It includes 4K QA pairs for 20 types of tasks. (4) *TempCompass* [33] assesses fine-grained temporal understanding with 410 short videos. We use all multi-choice QA tasks except for the video captioning task. (5) *EgoSchema* [35] features 5K egocentric video clips, each approximately 3 minutes long, with temporally demanding QA pairs. (6) *VideoMME* [13] is a general video QA benchmark covering diverse domains. It contains 2.7K QA samples over videos of varied lengths, ranging from 11 seconds to 1 hour. We use the long video split for evaluation.

**Implementation details.** Unless otherwise specified, we use Qwen2.5-VL-7B [4] as the base model. To strike a balance between training efficiency and memory consumption, we sample video frames at 2 FPS and adaptively resize each video input to contain approximately 2.8 million pixels. For instance, a 50-second video yields 100 frames, each with a resolution of roughly 96×96×3. During the reinforcement fine-tuning phase, we train for 5 epochs using a batch size of 8 and select the final checkpoint for evaluation. For fine-tuning on downstream benchmarks, we train for 2 epochs. More implementation details are provided in Appendix B.

Table 1: Performance of temporal video grounding on Charades-STA, ActivityNet, and TVGBench. The methods marked in gray* represent fine-tuning on corresponding benchmarks, while those in black indicate zero-shot settings. We compare our Time-R1 against existing 7B open-source LVLMs, as well as state-of-the-art VLP models.

| Type | Method | Charades-STA | | | ActivityNet | | | TVGBench | | |
|---|---|---|---|---|---|---|---|---|---|---|
| | | R1@0.3 | R1@0.5 | R1@0.7 | R1@0.3 | R1@0.5 | R1@0.7 | R1@0.3 | R1@0.5 | R1@0.7 |
| VLP | 2D-TAN* [68] | 57.3 | 45.8 | 27.9 | 60.4 | 43.4 | 25.0 | - | - | - |
| | UniVTG* [31] | 72.6 | 60.2 | 38.6 | 56.1 | 43.4 | 24.3 | - | - | - |
| | SSRN* [71] | - | 65.5 | 42.6 | - | 54.5 | 33.2 | - | - | - |
| | SnAG* [39] | - | 64.6 | 46.2 | - | 48.6 | 30.6 | - | - | - |
| | EaTR* [23] | - | 68.4 | 44.9 | - | 58.2 | 37.6 | - | - | - |
| | Gemini-2.5-Pro [10] | - | - | - | - | - | - | 39.1 | 24.4 | 12.8 |
| SFT | Momentor [43] | 42.6 | 26.6 | 11.6 | 42.9 | 23.0 | 12.4 | - | - | - |
| | ChatVTG [44] | 52.7 | 33.0 | 15.9 | 40.7 | 22.5 | 9.4 | - | - | - |
| | TimeChat [47] | - | 32.2 | 13.4 | 36.2 | 20.2 | 9.5 | 22.4 | 11.9 | 5.3 |
| | VTG-LLM [18] | - | 33.8 | 15.7 | - | - | - | - | - | - |
| | HawkEye [53] | 50.6 | 31.4 | 14.5 | 49.1 | 29.3 | 10.7 | - | - | - |
| | VTimeLLM [22] | 51.0 | 27.5 | 11.4 | 44.0 | 27.8 | 14.3 | - | - | - |
| | VideoChat-TPO [57] | 58.3 | 40.2 | 18.4 | - | - | - | - | - | - |
| | VideoExpert [70] | 61.5 | 40.3 | 20.9 | - | - | - | - | - | - |
| | TimeSuite [65] | 69.9 | 48.7 | 24.0 | - | - | - | 31.1 | 18.0 | 8.9 |
| | VideoMind [32] | 73.5 | 59.1 | 31.2 | 48.4 | 30.3 | 15.7 | - | - | - |
| | VideoChat-Flash [28] | 74.5 | 53.1 | 27.6 | - | - | - | 32.8 | 19.8 | 10.4 |
| | TRACE [19] | - | 40.3 | 19.4 | - | 37.7 | 24.0 | 37.0 | 25.5 | 14.6 |
| | HawkEye* [53] | 72.5 | 58.3 | 28.8 | 55.9 | 34.7 | 17.9 | - | - | - |
| | TimeSuite* [65] | 79.4 | 67.1 | 43.0 | - | - | - | - | - | - |
| | VideoChat-TPO* [57] | 77.0 | 65.0 | 40.7 | - | - | - | - | - | - |
| | VideoExpert* [70] | 74.3 | 60.8 | 36.5 | - | - | - | - | - | - |
| RL | Time-R1 (ours) | **78.1** | **60.8** | **35.3** | **58.6** | **39.0** | **21.4** | **41.8** | **29.4** | **16.4** |
| | Time-R1 (ours)* | 82.8 | 72.2 | 50.1 | 73.3 | 55.6 | 34.0 | - | - | - |

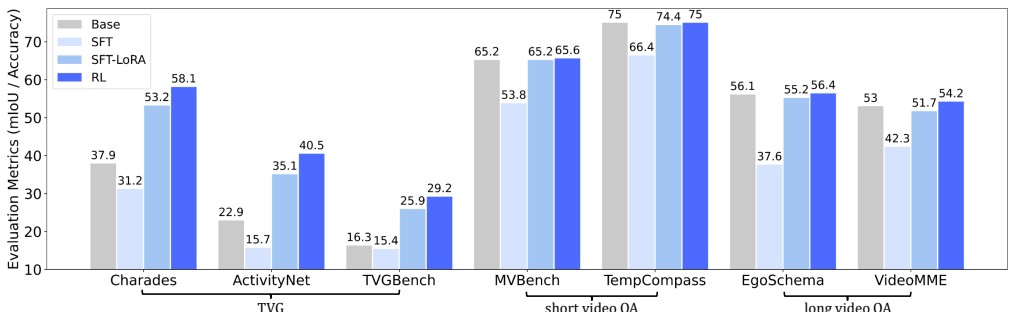

Figure 4: Comparison between post-training paradigms across various tasks, including short video QA, long video QA and temporal video grounding. Both "SFT" and "RL" full-finetune the LLM, while "SFT-LoRA" denotes finetuning the LLM with LoRA [21]. The "Base" is Qwen2.5-VL-7B.

**Evaluation metrics.** For TVG, following [47, 65], we adopt the "R1@m" evaluation protocol to compare with state-of-the-art models, which computes the percentage of samples where the top-1 predicted segment has an IoU greater than a threshold $m$, with $m \in \{0.3, 0.5, 0.7\}$. For brevity, we also adopt mIoU, which calculates the average IoU on all testing data as an alternative metric. For video QA, we report accuracy as the evaluation metric.

### 4.2 Comparison with State-of-the-Art

We compare Time-R1 with state-of-the-art TVG methods, including both traditional video-language pre-training models (VLP) and recent large video-language models fine-tuned via SFT.

**Time-R1 surpasses SFT-based models in zero-shot settings.** As shown in Table 1, in the zero-shot setting, Time-R1 demonstrates strong performance, outperforming SFT-based models that rely on large-scale supervision. Despite using only 2.5K samples for RL, Time-R1 achieves leading results across multiple benchmarks. On Charades-STA, Time-R1 attains an R1@0.3 score of 78.1, surpassing a range of strong SFT models such as VideoChat-Flash (74.5), VideoMind (73.5), and TimeSuite

(69.9). On ActivityNet, its R1@0.3 score reaches 58.6, which is also superior to other leading methods like VideoMind (48.4) and HawkEye (49.1). On our proposed TVGBench, its R1@0.3 score of 41.8 significantly exceeds all baselines, including TRACE (37.0) and the powerful closed-source model Gemini-2.5-Pro (39.1). These results highlight the data efficiency and strong generalization capabilities of our RL-based post-training approach.

**Time-R1\* outperforms all SFT-based LVLMs and many traditional VLP-based models.** When fine-tuned on downstream benchmarks, Time-R1\* consistently outperforms both traditional VLP-based and SFT-based models on the TVG task. On the Charades-STA dataset, its R1@0.7 score reaches 50.1 and its R1@0.5 score reaches 72.2, comprehensively outperforming all other methods, including highly competitive models like VideoChat-TPO\* (40.7, 65.0) and TimeSuite\* (43.0, 67.1). Notably, this superior performance is achieved using far fewer samples than the 349K SFT examples used by TimeSuite\*. On the more challenging ActivityNet dataset, Time-R1\* also achieves state-of-the-art results, with an R1@0.5 score of 55.6 that is superior to fine-tuned SFT models like TRACE (37.7) and classic VLP models like SSRN\* (54.5). This indicates that our framework not only generalizes well but also achieves exceptional performance through task-specific adaptation.

## 4.3 Ablation Study

We conduct a series of ablation studies to validate our key design choices. We first analyze our training strategies and the framework's generality, and conclude with an analysis of our reward design. More extensive experiments and detailed results are available in Appendix D.

**Utility of TimeRFT and Time-R1 components.** As shown in Table 2, we analyze the contributions of various training components within our framework. We observe that both Gaussian Filtering (GF) and Multi-Epoch training (ME) individually improve performance compared to the baseline (Row 1). Multi-Epoch training (Row 4) provides a particularly substantial

Table 2: Ablation of Time-R1-7B trainning. GF, ME, SF refers to Gaussian Filtering, Multi-Epoch, and Sample Filtering per epoch, respectively.

|  | tIoU | GF | ME | SF | TVGBench | | |
|---|---|---|---|---|---|---|---|
|  |  |  |  |  | R1@0.3 | R1@0.5 | R1@0.7 |
| 1 | ✗ | ✗ | ✗ | ✗ | 38.0 | 24.8 | 13.2 |
| 2 | ✓ | ✗ | ✗ | ✗ | 36.0 | 23.6 | 12.9 |
| 3 | ✗ | ✓ | ✗ | ✗ | 37.2 | 25.0 | 13.4 |
| 4 | ✗ | ✗ | ✓ | ✗ | 39.9 | 26.0 | 14.2 |
| 5 | ✓ | ✓ | ✗ | ✗ | 38.4 | 25.6 | 14.1 |
| 6 | ✓ | ✗ | ✓ | ✗ | 39.4 | 26.5 | 16.4 |
| 7 | ✓ | ✓ | ✓ | ✗ | 41.6 | 28.5 | 15.6 |
| 8 | ✓ | ✓ | ✓ | ✓ | 41.8 | 29.4 | 16.4 |

gain, lifting the R1@0.7 score from 13.2 to 14.2. Notably, the value of tIoU supervision becomes most prominent when combined with multi-epoch training. This combination (Row 6) leads to a significant leap across all metrics. It boosts the R1@0.7 performance substantially to 16.4. The progressive integration of these strategies culminates in our full model. The final addition of Sample Filtering (SF) in Row 8 further refines the results. This achieves our best performance with an R1@0.5 of 29.4 and an R1@0.7 of 16.4.

**Generalization of RL vs. SFT.** As shown in Figure 4, full fine-tuning with SFT on a small dataset significantly degrades generalization across all tasks, whereas RL consistently preserves it. While LoRA-based fine-tuning (SFT-LoRA) alleviates this issue, RL still demonstrates stronger overall performance and generalization. This advantage is particularly evident in data efficiency. Our RL model trained on only 2.5K samples consistently outperforms an SFT-LoRA model trained on a massive 339K dataset, as detailed in our appendix. For example, on ActivityNet, RL improves the mIoU from 16.3 to 29.2, while SFT-LoRA with the small dataset only reaches 25.9. Furthermore, RL also boosts performance on video QA benchmarks. On VideoMME, it increases performance from 53.0 to 54.2, while SFT-LoRA causes a decline to 51.7.

**Impact of cold start.** As shown in Figure 5, the impact of a cold start is particularly pronounced for the smaller 3B model, significantly boosting its mIoU performance from 18.0 to 20.3. While the larger 7B model shows only

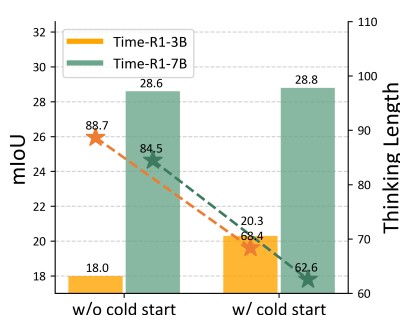

Figure 5: Impact of SFT-based cold start on IoU performance and thinking token count, with token counts marked by ⋆ on dashed lines.

a marginal improvement in mIoU, the cold start provides a notable efficiency benefit for both models. It reduces the thinking token length of the 3B model from 88.7 to 68.4 and the 7B model from 84.5 to 62.6. We attribute these effects to the cold start's function in suppressing hallucinations, a tendency more prevalent in weaker models.

**Effectiveness of the Time-R1 framework on different base models.** To demonstrate that our framework is broadly effective and not limited to a specific base model, we apply Time-R1 post-training to a diverse set of LVLMs, including the Qwen-2.5-VL [4], MiMo-VL [56], and InternVL3 [72]. As shown in Table 3, our approach consistently yields substantial performance gains across various model architectures and sizes on TVGBench. The improvements are particularly striking for models like MiMo-VL-7B, where the R1@0.7 score is more than doubled from 6.6 to 15.1. Similar significant boosts are observed for both smaller and larger models, with Qwen-2.5-VL-3B improving from

Table 3: Ablation on TVGBench across different base models and sizes.

| Model | Type | R1@0.3 | R1@0.5 | R1@0.7 |
|---|---|---|---|---|
| Qwen-2.5-VL-3B [4] | Base | 11.5 | 6.5 | 3.8 |
| | Time-R1 | **33.5** | **21.0** | **10.5** |
| Qwen-2.5-VL-7B [4] | Base | 24.9 | 16.0 | 8.0 |
| | Time-R1 | **41.6** | **28.5** | **15.6** |
| MiMo-VL-7B [56] | Base | 22.4 | 12.6 | 6.6 |
| | Time-R1 | **41.2** | **27.8** | **15.1** |
| InternVL3-2B [72] | Base | 16.3 | 6.3 | 2.3 |
| | Time-R1 | **21.8** | **9.5** | **4.1** |
| InternVL3-8B [72] | Base | 17.4 | 8.3 | 3.4 |
| | Time-R1 | **38.0** | **22.5** | **9.2** |

3.8 to 10.5 and InternVL3-8B from 3.4 to 9.2. This consistent effectiveness across different model families and sizes demonstrates the versatility and general applicability of our RL-based paradigm.

**Ablation on reward designs.** As shown in Table 4, we compare the timestamp-aware IoU reward ($r_{\text{tIoU}}$) with several alternatives, including standard IoU ($r_{\text{IoU}}$), sparse exact-match reward ($r_{\text{em}}$), distance-based metrics (absolute error $r_{\text{abs}}$ and RMSE $r_{\text{rmse}}$), and a variant using only the center-alignment term (the second term) of the tIoU ($r_{\text{center}}$). The results indicate that strong, fine-grained localization signals are necessary, as sparse rewards (e.g., $r_{\text{em}}$) or format-only constraints ($r_{\text{format}}$ only) are insufficient. While distance-based metrics perform reasonably, IoU-based designs provide more stable and

Table 4: Ablation on different reward designs on TVGBench.

| Reward Design | R1@0.3 | R1@0.5 | R1@0.7 |
|---|---|---|---|
| $r_{\text{tIoU}} + r_{\text{format}}$ (Ours) | **41.8** | **29.4** | **16.4** |
| $r_{\text{format}}$ only | 27.1 | 18.0 | 10.1 |
| $r_{\text{tIoU}}$ (w/o format) | 40.5 | 27.6 | 15.4 |
| $r_{\text{IoU}} + r_{\text{format}}$ | 41.4 | 28.0 | 15.8 |
| $r_{\text{em}} + r_{\text{format}}$ | 26.5 | 16.8 | 9.1 |
| $r_{\text{abs}} + r_{\text{format}}$ | 39.1 | 27.8 | 14.8 |
| $r_{\text{rmse}} + r_{\text{format}}$ | 38.9 | 27.0 | 15.8 |
| $r_{\text{center}} + r_{\text{format}}$ | 37.6 | 25.9 | 15.0 |

informative learning signals for temporal localization. Among these, our $r_{\text{tIoU}}$ outperforms $r_{\text{IoU}}$ across all metrics. When combine with the $r_{\text{format}}$, the full reward ($r_{\text{tIoU}} + r_{\text{format}}$) achieves the best overall performance, reaching 16.4 R1@0.7, surpassing the $r_{\text{IoU}} + r_{\text{format}}$ baseline (15.8) and all other alternatives. This confirms the superiority of our final reward design in precise temporal alignment and structured reasoning guidance.

# 5 Conclusion

In this work, we present Time-R1, a reinforcement learning based post-training framework that significantly improves the generalization of Large Vision-Language Models for Temporal Video Grounding. Unlike prior methods that rely on large-scale supervised fine-tuning, Time-R1 leverages a verifiable reward signal to unlock strong temporal reasoning from pretrained models using limited data. Our contributions include: (1) Time-R1, a reasoning-guided post-training framework that enhances TVG via RL; (2) TimeRFT, a curated dataset and training strategy that fosters temporal grounding; (3) TVGBench, a small yet comprehensive benchmark for evaluating LVLMs on TVG. Extensive experiments show that Time-R1 achieves SoTA performance across TVG benchmarks in both zero-shot and fine-tuned settings, surpassing prior LVLMs and traditional VLP-based models, while also improving general video understanding. We hope this work inspires future directions in data-efficient and generalizable video-language understanding via reinforcement learning.

**Limitations.** Our Time-R1 framework currently struggles to process ultra-long videos. We plan to address this long-horizon context challenge in future work.

## Acknowledgements

This work was partially supported by the Beijing Natural Science Foundation (No. L233008).

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

# A  Limitations

Despite achieving notable improvements on the TVG task, our approach still has several limitations. First, Time-R1 suffers from slower training and inference speeds, primarily due to its large model size and reliance on autoregressive text generation. Second, to manage GPU memory consumption, we use a relatively low frame sampling rate, which may result in the loss of fine-grained motion information across frames. Finally, Time-R1 currently cannot handle ultra-long videos, limiting its applicability in scenarios such as full-length movie understanding.

# B  Implementation Details

**Details of Time-R1 framework.** Inspired by DAPO [61], we adopt its token-level loss for training, rather than the sample-level loss used in GRPO. Apart from minor changes to the loss, all setting is identical to GRPO. Besides, we find that other techniques introduced in DAPO do not benefit the TVG task, thus aborting other techniques. We full-finetune the LLM parameters at every step, thus $\frac{\pi_\theta(o_i)}{\pi_{\theta_{\text{old}}}(o_i)} = 1$. The sample number $G$ is set to 8. The coefficient $\beta$ is set to 0.04.

**Details of TimeRFT training.** For RFT data filtering, we use a Gaussian distribution with a fixed variance of 0.2, while varying the mean to control sample selection. In our cold start phase, we construct 150 samples from our training data sources (e.g., YT-Temporal [59]) to fine-tune the LLM using LoRA [21], with a LoRA rank of 64 and a LoRA alpha of 128. All of our results are reported based on the final training epoch. For RL, we use a learning rate of 1e-6 with the AdamW optimizer with $\beta_1$=0.9, $\beta_2 = 0.999$, and a linear scheduler to decay the learning rate from 1e-6 to 0. We use a batch size of 8 with gradient accumulation set to 2.

**Details of our evaluation prompts.** As shown in Figure 11, for temporal video grounding, the prompts used for training and testing are designed to encourage the model to reason before responding, following a template-based answer format. For VideoQA, we have two versions of prompts: one with CoT and one without CoT.

**Details of TVG baseline methods and implementations.** We evaluate the baselines on TVGBench using their original best-performing setting, focusing primarily on video input and prompt design.

• TimeChat [47] is built upon the InstructBLIP [8] architecture and introduces a video Q-former to encode video tokens. It operates at a resolution of 224 and samples 96 frames.
• TRACE [19] treats each combination of timestamp, saliency score, and caption as a discrete event and enables the LVLM to autoregressively generate event sequences. It operates at a higher resolution of 336 and samples 128 frames.
• TimeSuite [65] introduces a token shuffling strategy to compress long video token sequences and incorporates positional encoding to enhance visual understanding. It adopts a resolution of 224 and samples 128 frames.
• VideoChat-Flash [28] proposes a progressive visual token dropping mechanism within intermediate LLM layers to compress video inputs and extend the effective context length. It uses a resolution of 448 and samples video at 1 fps, with a maximum of 512 frames.
• Gemini-2.5-Pro [10]: Gemini-2.5-Pro is state-of-the-art video understanding model capable of reasoning over videos exceeding one hour in length. It supports video question answering and temporal localization tasks.

Table 5: Comparison of different approaches on TVGBench for all types. We use mIoU as metric.

| Method | EC | ES | HAC | HAP | HAS | HP | OA | OC | OEC | OES | OT |
|---|---|---|---|---|---|---|---|---|---|---|---|
| TimeChat [47] | 22.3 | 32.8 | 16.6 | 9.8 | 14.6 | 35.1 | 15.0 | 9.2 | 2.4 | 18.0 | 10.2 |
| TimeSuite [65] | 27.3 | 39.6 | 14.2 | 12.8 | 24.9 | 39.6 | 14.6 | 13.9 | 6.7 | 32.6 | 14.3 |
| TRACE [19] | 57.1 | 66.8 | 25.9 | 17.5 | 26.5 | 45.1 | 17.8 | 22.1 | 12.5 | 36.8 | 24.9 |
| VideoChat-Flash [28] | 38.3 | 47.2 | 12.9 | 13.9 | 27.1 | 39.4 | 14.9 | 12.7 | 6.5 | 24.3 | 12.9 |
| Gemini-2.5-Pro [10] | 46.7 | 45.3 | 21.1 | 27.6 | 30.9 | 39.9 | 23.0 | 31.1 | 14.1 | 35.9 | 17.8 |
| Time-R1 (ours) | 49.3 | 65.3 | 28.3 | 24.3 | 39.3 | 56.2 | 26.3 | 21.8 | 9.0 | 32.7 | 21.8 |

**Details of our implemented SFT baselines.** We implemented two versions of SFT fine-tuning: one is full-parameter fine-tuning of the LLM (SFT), and the other is LoRA-based fine-tuning of the LLM (SFT-LoRA). For SFT-LoRA, the LoRA rank is set to 64, and the LoRA alpha is set to 128. Both configurations use the following settings: a learning rate of 2e-5, the AdamW optimizer with $\beta_1$=0.9, $\beta_2$ = 0.999, a weight decay of 0, the batch size of 8, and accumulation steps of 2. We fine-tune for 5 epochs on our 2.5K data, and use a linear scheduler to gradually decay the learning rate to 0.

## C  Additional Analyses

**In-depth comparisons of different approaches on TVGBench by semantic type.** Table 5 provides a detailed performance comparison of various methods on the TVGBench across different semantic categories. Specifically, the abbreviations represent: EC (Environment Change), ES (Environment State), HAC (Human Action – Complex), HAP (Human Action – Procedural), HAS (Human Action – Simple), HP (Human Pose), OA (Object Attribute), OC (Object Counting), OEC (Object Existence – Complex), OES (Object Existence – Simple), and OT (Object Transition). Detailed definition and construction process can be found in Figure 15.

Time-R1 demonstrates strong competitiveness across multiple semantic categories. First, particularly in the four tasks of HAC, HAS, HP and OA, Time-R1 achieved the highest scores among all compared methods, showcasing its excellent ability in understanding the details of human actions and identifying object features. For example, Time-R1 achieves an mIoU of 56.2 on HP, which is 11.1 points higher than the second-best method, TRACE with mIoU of 45.1. On HAS, Time-R1 reaches 39.3, outperforming Gemini-2.5-Pro's 30.9 by 8.4 points. Second, in the three tasks of ES, EC and OT, Time-R1 demonstrates strong performance comparable to the top model TRACE, with its performance being very close or immediately following. In the HAP task, Time-R1 also performs excellently, with its performance being in the same tier as Gemini-2.5-Pro. Last, all models still show a noticeable gap compared to Gemini in understanding complex instructions such as in HAP, OC, and OEC. For example, in HAP, which involves procedural activity localization, Gemini achieves 27.6, while our model ranks second with a score of 24.3. In object counting, Gemini attains 31.1, substantially outperforming our model's 21.8. In summary, Time-R1 performs well on both non-human simple instructions and human-related instructions, but there is still room for improvement in complex instruction grounding and object-related grounding.

Table 6: Inference speed comparison between HuggingFace Transformers and vLLM libraries. Speeds are reported as (with CoT / without CoT) with 8 GPUs.

| | TVGBench | Charades | ActivityNet |
|---|---|---|---|
| Domain | Mixed | Indoor | Human Activity |
| Sample Count | 800 | 3,720 | 17,031 |
| vLLM Speed | 8.3 / 6.9 min | 11.7 / 11.2 min | 36.1 / 33.2 min |
| Transformers Speed | 42.0 / 27.3 min | 3.3 / 1.2 hour | 15.0 / 7.5 hour |

**Comparison of speed and accuracy between inference library transformers and vLLM.** We observe that the inference speed of the implementation in the transformers [54] library is very slow. To address this, we implemented an accelerated inference version using vLLM [24] for all related 7 downstream benchmarks. As detailed in Table 6, vLLM delivers substantial performance gains across all datasets. For example, on the TVGBench benchmark, inference time with CoT is reduced from 42 minutes to just 8.3 minutes, achieving over a $5\times$ speedup.

Table 7: Ablation of data filtering strategies.

| Method | R1@0.3 | R1@0.5 | R1@0.7 | mIoU |
|---|---|---|---|---|
| random | 39.4 | 26.5 | 16.4 | 27.4 |
| gaussian (0.3) | 41.6 | 28.5 | 15.6 | 28.6 |
| gaussian (0.5) | 40.6 | 28.2 | 16.0 | 28.3 |
| gaussian (0.7) | 37.2 | 26.9 | 15.5 | 26.5 |
| uniform | 40.4 | 28.5 | 15.9 | 28.3 |

Table 8: Ablation of KL and CoT in GRPO.

| KL | CoT | R1@0.3 | R1@0.5 | R1@0.7 | mIoU |
|---|---|---|---|---|---|
| ✗ | ✗ | 40.4 | 29.1 | 14.9 | 28.1 |
| ✓ | ✗ | 40.8 | 27.4 | 15.0 | 27.7 |
| ✗ | ✓ | 42.9 | 29.5 | 15.0 | 29.1 |
| ✓ | ✓ | 41.6 | 28.5 | 15.6 | 28.6 |

Table 9: Comparison of the token-level loss design used by DAPO [61] and the sample-level loss design used by GRPO [48].

| Loss | Charades-STA | | | | ActivityNet | | | | TVGBench | | | |
|---|---|---|---|---|---|---|---|---|---|---|---|---|
| | R1@0.3 | R1@0.5 | R1@0.7 | mIoU | R1@0.3 | R1@0.5 | R1@0.7 | mIoU | R1@0.3 | R1@0.5 | R1@0.7 | mIoU |
| GRPO | 76.7 | 59.8 | 34.4 | 57.0 | 55.9 | 37.1 | 20.3 | 37.8 | 40.8 | 28.0 | 16.5 | 28.4 |
| DAPO | 77.4 | 60.0 | 34.1 | 57.2 | 56.2 | 37.4 | 20.4 | 38.0 | 41.6 | 28.5 | 15.6 | 28.6 |

# D  Ablation Studies

**Ablation of different RFT data filtering strategies.** As shown in Table 7, different data filtering strategy in the initial round affects model's performance. First, appropriate Gaussian filtering outperforms both uniform and random filtering methods. Among the Gaussian filtering settings, a standard deviation of 0.3 yields the best results, followed by 0.5 and then 0.7. These findings suggest that incorporating moderately challenging samples during RFT helps improve the model's generalization capability more effectively than using either overly easy or extremely difficult examples.

**Ablation of KL and CoT during GRPO training.** As shown in Table 8, incorporating CoT reasoning during training leads to improved performance compared to the No-CoT setting, suggesting that CoT enhances the model's temporal video grounding capabilities. When KL divergence is omitted (No-KL), performance slightly decreases under the No-CoT setting but unexpectedly improves when CoT is present. However, we find that in the No-KL+CoT setting, the model often fails to produce thinking process, directly jumping to answers. In contrast, using KL divergence helps maintain more logical reasoning that is easier to follow. To balance performance and interpretability, we adopt a training setup that includes both KL and CoT.

**Comparison of tIoU and IoU during multi-epoch training.** As shown in Figure 6, tIoU consistently outperforms standard IoU during both the early and late stages of training over the first 5 epochs. Notably, while tIoU steadily improves as training progresses, IoU shows a decline in performance by the fifth epoch. This highlights the advantage of using tIoU as a more stable and reliable reward for temporal video grounding.

**Ablation of sample filtering in multi-epoch training.** As shown in Figure 7, applying sample filtering (SF) to remove simpler training samples yields consistent performance improvements across epochs. This suggests that easy samples with high-IoU may introduce noise or reduce the effectiveness of learning, and filtering them helps focus the model on more informative and challenging instances.

**Ablation of DAPO & GRPO.** The sample-level loss used by GRPO computes the loss by averaging over each individual sample. This approach leads to unequal loss contributions for tokens when dealing with CoTs of varying lengths. DAPO addresses this issue by employing a token-level loss. The underlying principle is that the token-level loss can effectively guide the model in the process of CoT generation, allowing it to learn useful patterns from CoTs of different lengths sampled during training. In Table 9, we compare these two loss designs. We empirically find that DAPO outperforms GRPO on the majority of metrics, thus we adopt DAPO's loss design.

Table 10: Performance comparison between the base model, RL, and different SFT settings.

| Method | TVGBench | | | | Charades | | | | ActivityNet | | | |
|---|---|---|---|---|---|---|---|---|---|---|---|---|
| | R1@0.3 | R1@0.5 | R1@0.7 | mIoU | R1@0.3 | R1@0.5 | R1@0.7 | mIoU | R1@0.3 | R1@0.5 | R1@0.7 | mIoU |
| Base Model | 24.9 | 16.0 | 8.0 | 16.3 | 58.7 | 38.3 | 16.6 | 37.9 | 34.3 | 21.6 | 12.9 | 22.9 |
| SFT-2.5K | 25.9 | 14.4 | 5.8 | 15.4 | 49.5 | 31.3 | 12.7 | 31.2 | 27.7 | 13.6 | 5.8 | 15.7 |
| SFT-LoRA-2.5K | 38.6 | 24.5 | 14.6 | 25.9 | 74.0 | 55.7 | 29.9 | 53.2 | 52.3 | 34.3 | 18.8 | 35.1 |
| SFT-LoRA-339K | 38.9 | 28.2 | 15.2 | 27.4 | 72.4 | 54.7 | 26.9 | 51.7 | 45.7 | 29.3 | 16.4 | 30.5 |
| Time-R1-2.5K | 41.8 | 29.4 | 16.4 | 29.2 | 78.1 | 60.8 | 35.3 | 58.1 | 58.1 | 39.0 | 21.4 | 40.5 |

Table 11: Ablation study on SFT-LoRA hyperparameters (rank/alpha).

| LoRA rank / alpha | TVGBench | | | | Charades | | | | ActivityNet | | | |
|---|---|---|---|---|---|---|---|---|---|---|---|---|
| | R1@0.3 | R1@0.5 | R1@0.7 | mIoU | R1@0.3 | R1@0.5 | R1@0.7 | mIoU | R1@0.3 | R1@0.5 | R1@0.7 | mIoU |
| 16 / 32 | 31.4 | 22.0 | 11.5 | 21.8 | 58.7 | 44.9 | 25.6 | 40.4 | - | - | - | - |
| 32 / 64 | 35.0 | 24.2 | 14.4 | 24.8 | 72.4 | 54.0 | 29.4 | 48.8 | - | - | - | - |
| 64 / 128 | 38.5 | 24.5 | 14.6 | 26.2 | 74.0 | 55.7 | 29.9 | 49.9 | 52.3 | 34.3 | 18.8 | 36.5 |
| 128 / 256 | 38.4 | 25.4 | 14.2 | 26.4 | 76.5 | 58.3 | 32.9 | 52.2 | - | - | - | - |
| 256 / 512 | 39.1 | 27.1 | 15.1 | 26.9 | 75.9 | 57.4 | 32.7 | 51.6 | - | - | - | - |
| Time-R1-7B | 41.6 | 28.5 | 15.6 | 28.6 | 77.4 | 60.0 | 34.1 | 57.2 | 58.1 | 39.0 | 21.4 | 40.5 |

Table 12: Ablation study on the number of training epochs for SFT-LoRA.

| LoRA rank / alpha | Epochs | TVGBench | | | | Charades | | | | ActivityNet | | | |
|---|---|---|---|---|---|---|---|---|---|---|---|---|---|
| | | R1@0.3 | R1@0.5 | R1@0.7 | mIoU | R1@0.3 | R1@0.5 | R1@0.7 | mIoU | R1@0.3 | R1@0.5 | R1@0.7 | mIoU |
| 64 / 128 | 1 | 38.0 | 25.1 | 14.1 | 26.6 | 73.5 | 57.3 | 33.3 | 50.5 | 54.8 | 37.1 | 21.7 | 39.0 |
| | 2 | 36.9 | 24.6 | 14.1 | 25.9 | 73.5 | 56.2 | 31.1 | 50.3 | 52.4 | 35.7 | 20.5 | 37.4 |
| | 5 | 38.5 | 24.5 | 14.6 | 26.2 | 74.0 | 55.7 | 29.9 | 49.9 | 52.3 | 34.3 | 18.8 | 36.5 |
| 128 / 256 | 1 | 27.4 | 18.1 | 8.9 | 18.3 | 47.8 | 33.4 | 16.7 | 31.7 | - | - | - | - |
| | 2 | 37.9 | 26.2 | 14.4 | 26.2 | 76.4 | 57.0 | 30.7 | 51.3 | - | - | - | - |
| | 5 | 38.4 | 25.4 | 14.2 | 26.4 | 76.5 | 58.3 | 32.9 | 52.2 | - | - | - | - |

**Comparison between RL and SFT.** We conduct a comprehensive comparison between our Reinforcement Learning (RL) approach and various Supervised Fine-Tuning (SFT) settings. As shown in Table 10, full-parameter SFT on our curated 2.5K dataset leads to severe overfitting and a significant performance drop compared to the base model. Using LoRA for parameter-efficient fine-tuning (SFT-LoRA) mitigates this issue and improves performance, but it still falls short of the results achieved with our RL method. To further demonstrate the data efficiency of our RL paradigm, we train an SFT-LoRA model on the full 339K TVG dataset. Even with access to over 100 times more data, the SFT-LoRA model underperforms our RL model, which is trained on only 2.5K samples. This highlights the superior generalization ability and data efficiency of the proposed RL framework.

Further ablation studies on SFT-LoRA are presented in Table 11 and Table 12. We find that while tuning hyperparameters like LoRA rank / alpha and the number of training epochs can lead to better SFT results, none of the tested configurations match the performance of our RL approach. These extensive comparisons firmly establish the advantages of our RL-based post-training framework over SFT for the TVG task.

**Ablation of cold-start strategy.** We investigate the effect of using a cold-start strategy, where we initialize the model with SFT on dense video captioning data before starting RL training. This approach is motivated by our empirical observation that the model naturally generates reasoning similar to video descriptions. The cold-start strategy helps accelerate convergence towards the desired response format and provides a better initialization for the RL policy. As shown in Table 13, this initialization provides a significant performance boost, particularly for smaller models like the 3B variant, improving its ability to learn effectively during the subsequent RL phase.

**Effectiveness of the Time-R1 framework on different base models.** To demonstrate the effectiveness of our Time-R1 framework, we apply it to a diverse set of Large Vision-Language Models with varying architectures and sizes. As shown in Table 14, our RL-based post-training consistently yields substantial performance improvements across all tested models, including Qwen-VL (3B, 7B), MiMo-VL-7B, and InternVL (2B, 8B). The results confirm that our method is not limited to a specific backbone and can effectively enhance the temporal grounding capabilities of different LVLMs. We also observe that larger models generally achieve better performance both before and after post-training, supporting the idea that scaling model capacity is beneficial for TVG tasks. The Time-R1* results indicate further fine-tuning on downstream datasets, which yields additional performance gains.

**Ablation of different reward designs.** We perform a comprehensive ablation study to validate our choice of the reward function. As shown in Table 15, We compare the timestamp-aware IoU ($r_{\text{tIoU}}$) combined with the reasoning format reward against several alternatives. For clarity, we first define each reward component used in our experiments.

• Reasoning format reward $r_{\text{format}}(\cdot)$. A binary reward that is 1 if the model's output adheres to the required reasoning template format, and 0 otherwise. This reward encourages the model to generate responses in the correct structure.

Table 13: Performance of the 3B model with and without the SFT-based cold-start strategy.

| Method | TVGBench | | | | Charades | | | | ActivityNet | | | |
|---|---|---|---|---|---|---|---|---|---|---|---|---|
| | R1@0.3 | R1@0.5 | R1@0.7 | mIoU | R1@0.3 | R1@0.5 | R1@0.7 | mIoU | R1@0.3 | R1@0.5 | R1@0.7 | mIoU |
| w/o Cold-Start | 29.0 | 17.1 | 8.0 | 18.0 | 67.0 | 44.9 | 21.5 | 44.4 | 36.6 | 18.8 | 7.8 | 21.1 |
| w/ Cold-Start | 33.5 | 21.0 | 10.5 | 21.7 | 74.6 | 53.1 | 26.0 | 51.2 | 40.0 | 21.0 | 8.7 | 23.2 |

Table 14: Performance comparison across different base models and sizes.

| Model | Method | Charades | | | | ActivityNet | | | | TVGBench | | | |
|---|---|---|---|---|---|---|---|---|---|---|---|---|---|
| | | R1@0.3 | R1@0.5 | R1@0.7 | mIoU | R1@0.3 | R1@0.5 | R1@0.7 | mIoU | R1@0.3 | R1@0.5 | R1@0.7 | mIoU |
| Qwen-2.5-VL-3B | Base | 24.2 | 15.5 | 8.1 | 16.3 | 13.0 | 7.1 | 3.3 | 9.8 | 11.5 | 6.5 | 3.8 | 8.3 |
| | Time-R1 | 74.6 | 53.1 | 26.0 | 51.2 | 40.0 | 21.0 | 8.7 | 23.2 | 33.5 | 21.0 | 10.5 | 21.7 |
| | Time-R1* | 78.7 | 64.1 | 36.9 | 59.9 | 66.8 | 46.8 | 24.7 | 46.1 | - | - | - | - |
| Qwen-2.5-VL-7B | Base | 58.7 | 38.3 | 16.6 | 37.9 | 34.3 | 21.6 | 12.9 | 22.9 | 24.9 | 16.0 | 8.0 | 16.3 |
| | Time-R1 | 78.1 | 60.8 | 35.5 | 58.1 | 58.1 | 39.0 | 21.4 | 40.5 | 41.8 | 29.4 | 16.4 | 29.2 |
| | Time-R1* | 82.8 | 72.2 | 50.1 | 60.9 | 73.3 | 55.6 | 34.0 | 52.1 | - | - | - | - |
| MiMo-VL-7B | Base | 48.5 | 27.0 | 12.1 | 31.7 | 31.3 | 19.7 | 12.1 | 24.2 | 22.4 | 12.6 | 6.6 | 15.7 |
| | Time-R1 | 79.9 | 63.9 | 33.4 | 53.9 | 45.6 | 27.2 | 14.2 | 31.9 | 41.2 | 27.8 | 15.1 | 27.4 |
| InternVL-2B | Base | 20.9 | 7.8 | 1.9 | 15.4 | 18.6 | 8.5 | 3.1 | 14.2 | 16.3 | 6.3 | 2.3 | 11.7 |
| | Time-R1 | 24.0 | 11.5 | 3.5 | 15.7 | 20.6 | 9.5 | 3.9 | 14.2 | 21.8 | 9.5 | 4.1 | 14.8 |
| InternVL-8B | Base | 27.8 | 11.9 | 3.7 | 20.6 | 33.1 | 18.4 | 10.3 | 24.0 | 17.4 | 8.3 | 3.4 | 11.8 |
| | Time-R1 | 70.0 | 45.1 | 18.3 | 44.1 | 46.8 | 25.9 | 11.7 | 31.1 | 38.0 | 22.5 | 9.2 | 24.2 |

• Standard IoU reward $r_{\mathrm{IoU}}(\cdot)$. The standard Intersection over Union between the predicted segment $[t_s, t_e]$ and the ground-truth segment $[t'_s, t'_e]$, computed as:

$$r_{\mathrm{IoU}} = \frac{\max(0, \min(t_e, t'_e) - \max(t_s, t'_s))}{\max(t_e, t'_e) - \min(t_s, t'_s)} \tag{8}$$

• Timestamp-aware IoU reward $r_{\mathrm{tIoU}}(\cdot)$. The timestamp-aware IoU reward augments the standard IoU with a center alignment term that penalizes discrepancies between the centers of the predicted and ground-truth segments:

$$r_{\mathrm{tIoU}} = r_{\mathrm{IoU}} + r_{\mathrm{center}}, \quad \text{where} \quad r_{\mathrm{center}} = 1 - \frac{|(t_s + t_e)/2 - (t'_s + t'_e)/2|}{t'_e - t'_s} \tag{9}$$

This modification provides a more fine-grained grounding signal by encouraging both boundary alignment and temporal center consistency.

• Exact matching reward $r_{\mathrm{em}}(\cdot)$. A sparse binary reward that is 1 only if the predicted timestamps exactly match the ground truth, and 0 otherwise:

$$r_{\mathrm{em}} = \mathbb{I}(t_s = t'_s \wedge t_e = t'_e) \tag{10}$$

• Absolute error reward $r_{\mathrm{abs}}(\cdot)$. The negative L1 distance between the predicted and ground-truth boundaries:

$$r_{\mathrm{abs}} = -(|t_s - t'_s| + |t_e - t'_e|) \tag{11}$$

• RMSE reward $r_{\mathrm{rmse}}(\cdot)$. The negative Root Mean Square Error, which penalizes larger boundary errors more heavily:

$$r_{\mathrm{rmse}} = -\sqrt{\frac{(t_s - t'_s)^2 + (t_e - t'_e)^2}{2}} \tag{12}$$

The results in Table 15 clearly demonstrate the necessity of a strong, fine-grained grounding signal combined with a structural incentive. Using the reasoning format reward alone ($r_{\mathrm{format}}$ only) or sparse rewards like $r_{\mathrm{em}}$ is insufficient for effective learning. The comparison between $r_{\mathrm{tIoU}}$ (without format) and our full approach ($r_{\mathrm{tIoU}} + r_{\mathrm{format}}$) shows that the format reward consistently improves performance by guiding the model's generation structure. While distance-based metrics perform reasonably well, IoU-based approaches provide a more informative and stable learning signal. Among these, our proposed $r_{\mathrm{tIoU}}$ consistently outperforms the standard $r_{\mathrm{IoU}}$, validating the superiority of our final reward design. The combination of a precise grounding signal and a structured reasoning incentive yields the best temporal grounding performance.

## E   Qualitative Result

**Case study of temporal video grounding on Charades and ActivityNet.** As shown in Figure 8, in the example above, given a relatively complex language instruction, Time-R1 demonstrates more

Table 15: Ablation study on different reward functions and their combinations across three datasets.

| Reward Design | TVGBench | | | | Charades | | | | ActivityNet | | | |
|---|---|---|---|---|---|---|---|---|---|---|---|---|
| | R1@0.3 | R1@0.5 | R1@0.7 | mIoU | R1@0.3 | R1@0.5 | R1@0.7 | mIoU | R1@0.3 | R1@0.5 | R1@0.7 | mIoU |
| $r_{\text{tIoU}} + r_{\text{format}}$ (Ours) | 41.8 | 29.4 | 16.4 | 29.2 | 78.1 | 60.8 | 35.3 | 58.1 | 58.1 | 39.0 | 21.4 | 40.5 |
| $r_{\text{format}}$ only | 27.1 | 18.0 | 10.1 | 18.8 | 60.4 | 39.2 | 17.4 | 39.3 | 35.1 | 22.2 | 13.1 | 26.0 |
| $r_{\text{tIoU}}$ (w/o format) | 40.5 | 27.6 | 15.4 | 28.0 | 78.4 | 61.6 | 36.1 | 54.0 | 55.2 | 36.8 | 20.2 | 38.4 |
| $r_{\text{IoU}} + r_{\text{format}}$ | 41.4 | 28.0 | 15.8 | 28.4 | 77.6 | 58.9 | 32.7 | 56.4 | 58.8 | 38.8 | 20.7 | 39.4 |
| $r_{\text{center}} + r_{\text{format}}$ | 37.6 | 25.9 | 15.0 | 26.3 | 75.6 | 57.7 | 31.9 | 51.3 | 52.5 | 34.1 | 18.5 | 36.6 |
| $r_{\text{em}} + r_{\text{format}}$ | 26.5 | 16.8 | 9.1 | 18.3 | 61.6 | 40.3 | 18.7 | 40.4 | 35.9 | 22.5 | 13.3 | 26.4 |
| $r_{\text{abs}} + r_{\text{format}}$ | 39.1 | 27.8 | 14.8 | 27.4 | 76.2 | 59.5 | 33.9 | 52.3 | 51.4 | 32.9 | 17.8 | 35.7 |
| $r_{\text{rmse}} + r_{\text{format}}$ | 38.9 | 27.0 | 15.8 | 27.2 | 75.4 | 58.6 | 33.1 | 51.5 | 51.2 | 32.7 | 17.8 | 35.5 |

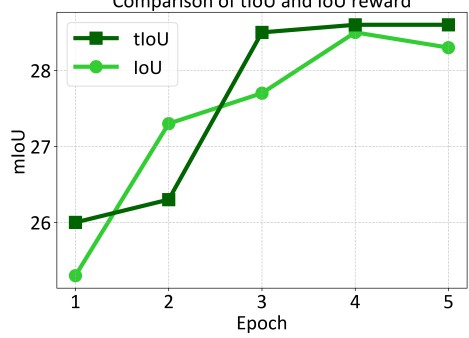

Figure 6: Performance comparison of tIoU and IoU in multi-epoch training.

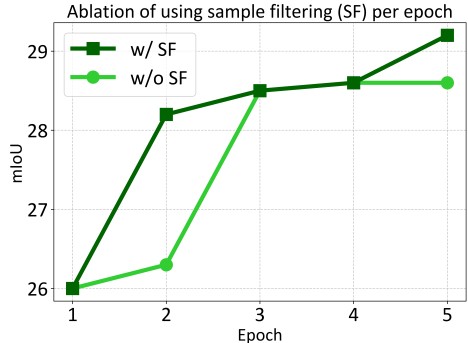

Figure 7: Ablation of sample filtering in multi-epoch training.

accurate localization than all baselines, successfully capturing the initial event "goes back to the pink bucket" within the timestamp, while other SoTA models like Gemini-2.5-Pro fail. In the example below, the model accurately localizes the event, excluding "a person is lying on the couch" and correctly distinguishing between sitting and lying, unlike other models which either localize only a small segment (TimeSuite and VideoChat-Flash) or the entire segment (TRACE and Gemini-2.5-Pro).

**Case study of short video QA on VideoMME and TempCompass.** As shown in Figures 9 and 10, Time-R1 demonstrates improved performance over the base model in tasks requiring positional judgment, scene storyline reasoning, and visual reasoning. For example, in Figure 9, Time-R1 correctly identifies that a car in the video is missing its right-front wheel, a detail that the base model fails to recognize. This reflects that Time-R1 likely possesses stronger video localization capabilities, which in turn enhance its visual reasoning ability. In Figure 12, we output a CoT when answering the QA task, providing some interpretability. This example shows that Time-R1's reasoning process is more concise, whereas the base model often reasons correctly but arrives at the wrong answer. This suggests that Time-R1's reasoning may be more effective in guiding the final answer, possibly benefiting from the outcome-driven RL of GRPO.

**Case study of long video QA on EgoSchema and VideoMME.** Figure 13 presents a long egocentric video QA example focused on summarizing task steps. In the "Hanging the Dress" case, the base model fails to identify all key steps, while our Time-R1 model correctly selects the answer by generating a more accurate chain-of-thought (CoT). In Figure 14, the task involves identifying the primary tools and ingredients used in the video and understanding their respective contributions. An additional example in Figure 9 involves animated scene reasoning, where Time-R1 correctly infers—based on elements like the presence of a guitar—that the heroes defeated the enemies using the power of music, whereas the base model misinterprets the cause. Across these complex reasoning tasks involving long videos, Time-R1 consistently demonstrates superior performance compared to the base model.

**Illustration of our prompt at training and inference time.** Figure 11 presents the prompts used for the temporal video grounding and video QA tasks at both training and inference time.

**Illustration of our prompt to annotate query semantics on TVGBench.** Figure 15 presents the prompt used to annotate query semantics in the TVGBench dataset. The prompt is designed to guide

the DeepSeek LLM in classifying each input query into one of 11 predefined semantic categories. To improve annotation quality, we refer to the construction of existing benchmarks and carefully select both positive and negative examples for each semantic type. These examples are chosen to include queries that are either easy or difficult for the model to answer, helping to refine the model's understanding and improve labeling accuracy.

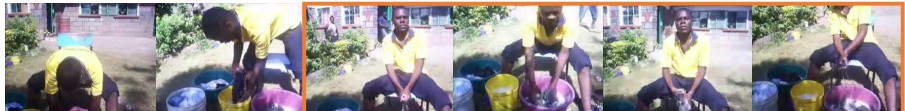

**Question**: Where is the event "The man goes back to the pink bucket to wash another dark item clothing, he wrings it out in the pink bucket, throws it in the yellow bucket, then wrings his hands together a few times while staring at the camera"?

(video length: 74s)

| | |
|---|---|
| Ground Truth | 33.1s ├──────────────────────────────────┤ 73.5s |
| TimeSuite | 20.0s ├────┤ 30.0s |
| Gemini-2.5-Pro | 57.0s ├──────────────┤ 73.0s |
| VideoChat-Flash | 0.0s ├──┤ 17.2s |
| Trace | 54.9s ├──────────────┤ 74.1s |
| Time-R1 (ours) | 35.0s ├──────────────────────────────────┤ 75.0s |

**Time-R1 thinking (ours):** The event "The man goes back to the pink bucket to wash another dark item clothing, he wrings it out in the pink bucket, throws it in the yellow bucket, then wrings his hands together a few times while staring at the camera." occurs between 35.00 and 75.00 seconds in the video.

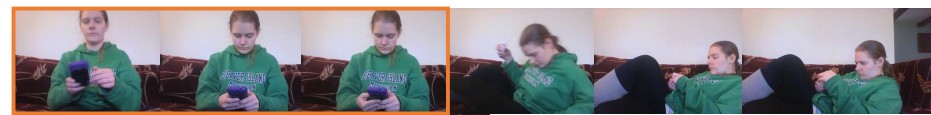

**Question**: Where is the event "a person is seen sitting on a couch"?

(video length: 31s)

| | |
|---|---|
| Ground Truth | 0.0s ├──────────────┤ 12.8s |
| TimeSuite | 0.0s ├──┤ 5.0s |
| Gemini-2.5-Pro | 0.0s ├────────────────────────────────┤ 30.0s |
| VideoChat-Flash | 0.0s ├──┤ 5.0s |
| Trace | 0.0s ├────────────────────────────────┤ 31.6s |
| Time-R1 (ours) | 0.0s ├──────────────┤ 10.0s |

**Time-R1 thinking (ours):** The event "a person is seen sitting on a couch" occurs at the beginning of the video. The person is seated on a couch, holding a phone, and appears to be engaged with it. This aligns with the initial frames of the video.

Figure 8: Temporal video grounding cases from Charades and ActivityNet benchmark. Time-R1 outperforms other SoTA models.

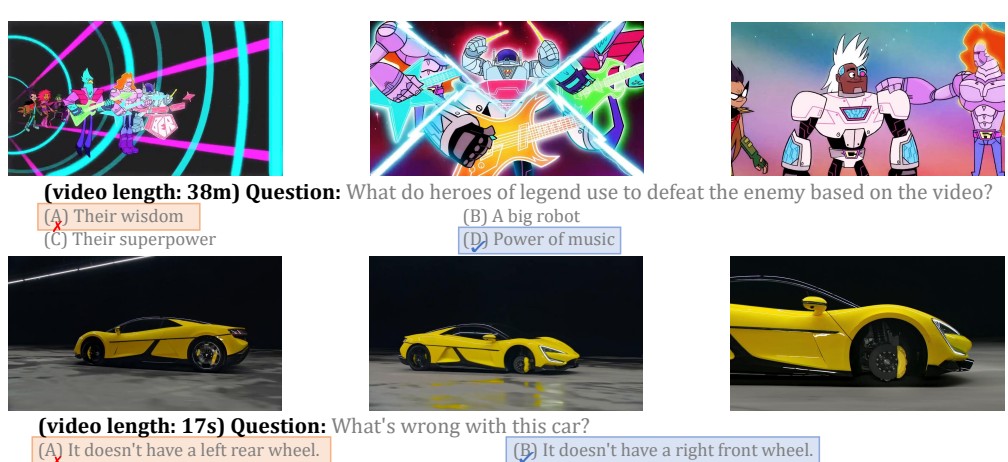

**(video length: 38m) Question:** What do heroes of legend use to defeat the enemy based on the video?
(A) Their wisdom
(B) A big robot
(C) Their superpower
(D) Power of music

**(video length: 17s) Question:** What's wrong with this car?
(A) It doesn't have a left rear wheel.
(B) It doesn't have a right front wheel.
(C) Its headlamp is broken.
(D) Its right door is broken.

Figure 9: Case study on VideoMME (w/o CoT), demonstrating that Time-R1 achieves better performance than the base model.

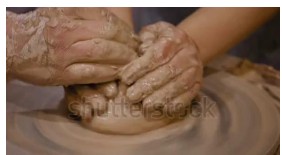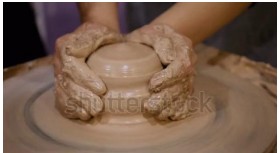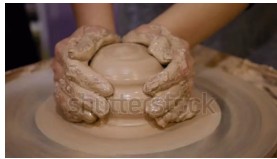

**(video length: 10s)Question:** Which sentence better captures the essence of the video?

(A) First, two hand are holding the clay pot and then three hands are holding it.

(B) In the video, three hands are holding the clay pot, then two hands are holding the clay pot.

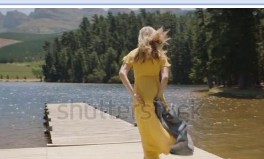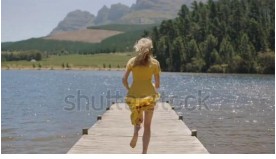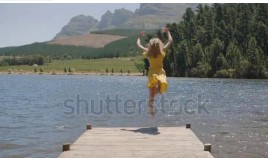

**(video length: 14s)Question:** What is the sequence of actions performed by the girl?

(A) Putting on clothes, jumping into water, taking off clothes

(B) Jumping into water, putting on clothes, taking off clothes

(C) Taking off clothes, putting on clothes, jumping into water

Figure 10: Case study on TempCompass (w/o CoT), demonstrating that Time-R1 achieves better performance than the base model.

---

### Temporal Video Grounding

**System Prompt**
You are a video analysis expert.

**Input Prompt**
To accurately pinpoint the event "[EVENT]" in the video, determine the precise time period of the event. Output your thought process within the <think> </think> tags, including analysis with either specific time ranges (xx.xx to xx.xx) in <timestep> </timestep> tags. Then, provide the start and end times (in seconds, precise to two decimal places) in the format "start time to end time" within the <answer> </answer> tags. For example: "12.54 to 17.83"."""

### Video QA

**System Prompt**
You are a video analysis expert.

**Input Prompt (w/o CoT)**
Carefully watch the video and pay attention to the cause and sequence of events, the detail and movement of objects and the action and pose of persons. Based on your observations, select the best option that accurately addresses the question. Provide the letter of the best option wrapped in parentheses within the <answer> </answer> tags. Question: {} {}""",

**Input Prompt (w/ CoT)**
Carefully watch the video and pay attention to the cause and sequence of events, the detail and movement of objects and the action and pose of persons. Based on your observations, select the best option that accurately addresses the question. Output your thought process within the <think> </think> tags, including analysis with either specific timestamps (xx.xx) or time ranges (xx.xx to xx.xx) in <timestep> </timestep> tags. Then, provide the letter of the best option wrapped in parentheses within the <answer> </answer> tags. Question: {} {}

Figure 11: Illustration of prompts at both training and inference time.

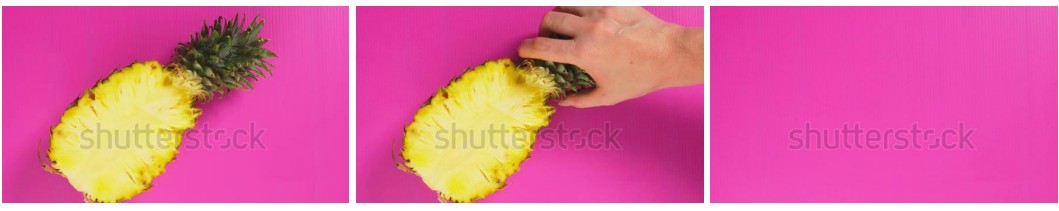

**(video length: 8s) Question:** Is the pineapple being pushed forward by a person?

(A) Yes 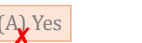 ✗

(B) No 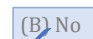 ✓

### Base model thinking:

To determine if the pineapple is being pushed forward by a person, let's analyze the video:

1. The video shows a pineapple on a pink surface.
2. A hand appears in the frame and interacts with the pineapple.
3. The hand lifts the pineapple, indicating that it is being moved by the person.

Given this analysis, the pineapple is indeed being pushed forward by a person.

### Time-R1 thinking (ours) :

The video shows a hand reaching into the frame and grasping the pineapple. The hand then lifts the pineapple, indicating that the pineapple is not being pushed forward by a person but rather being moved by the hand.

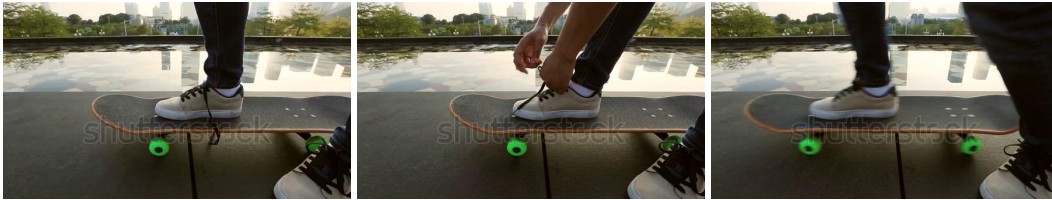

**(video length: 9s) Question:** In which sequence does the man perform the actions?

(A) Gliding on the skateboard and then jumping

(B) Jumping and then tying his shoelaces

(C) Tying his shoelaces and then gliding on the skateboard ✓

(D) Gliding on the skateboard and then tying his shoelaces ✗

### Base model thinking:

Let's analyze the sequence of actions performed by the man in the video:

1. The man is initially seen standing on a skateboard.
2. He then proceeds to tie his shoelaces.
3. After tying his shoelaces, he glides on the skateboard.

Given this sequence, the correct answer is: (D) gliding on the skateboard and then tying his shoelaces

### Time-R1 thinking (ours) :

The sequence of actions performed by the man in the video is as follows: first, he ties his shoelaces, and then he glides on the skateboard. This can be observed from the video where the man is seen tying his shoelaces before he starts gliding on the skateboard.

Figure 12: Case study on TempCompass (w/ CoT), demonstrating that Time-R1 achieves better performance than the base model.

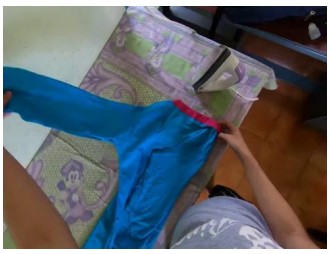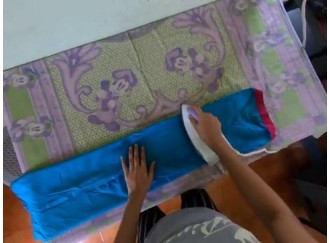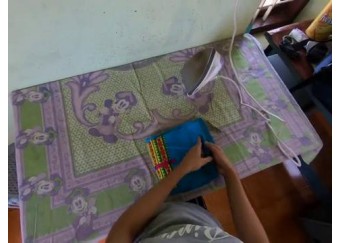

**(video length: 3m) Question:** Summarize the steps taken by c to complete the tasks in this video. don't list every action narrated, but rather focus on major components and how they're executed.

(A) C folds the dress, places it on the ironing board, and then hangs it up.

(B) C folds the dress, places it on the ironing board, and then packs it.

(C) C folds the dress, places it on the ironing board, and then washes it.

(D) C folds the dress, places it on the ironing board, and then irons it and hangs it up.

(E) C folds the dress, places it on the ironing board, irons it, and then folds it again.

## Base model thinking:

Let's analyze the video step by step to determine the correct sequence of actions:

1. Folding the Dress: The video starts with C folding a blue dress on the ironing board. This is the first major action.

2. Placing the Dress on the Ironing Board: After folding the dress, C places it on the ironing board.

3. Ironing the Dress: C then irons the dress, ensuring it is smooth and free of wrinkles.

4. **Hanging the Dress**: Finally, C hangs the dress up, completing the task.

Now, let's evaluate the options:

(A) C folds the dress, places it on the ironing board, and then hangs it up.

(B) C folds the dress, places it on the ironing board, and then packs it.

(C) C folds the dress, places it on the ironing board, and then washes it.

(D) C folds the dress, places it on the ironing board, and then irons it and hangs it up.

(E) C folds the dress, places it on the ironing board, irons it, and then folds it again.

The correct sequence of actions matches option (D).

## Time-R1 thinking (ours) :

The video shows C folding a dress, placing it on the ironing board, and then ironing it. The sequence of actions is clear and follows a logical order. The dress is folded, placed on the ironing board, and then ironed, which is a common sequence for completing a task like ironing a dress.

Figure 13: Case study on EgoSchema (w/ CoT), demonstrating that Time-R1 achieves better performance than the base model.

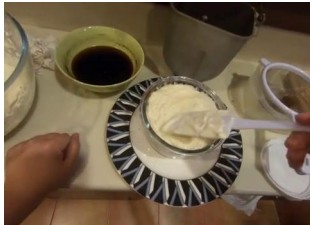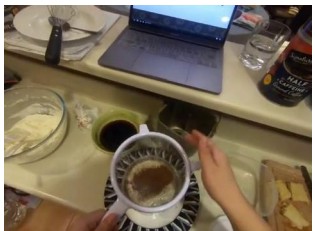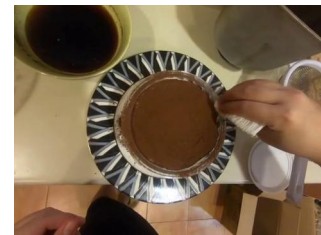

**(video length: 3m) Question:** Identify the primary tools and ingredients c used throughout the video, and discuss how they contributed to the overall dessert preparation process.

(A) The primary tools and ingredients c used throughout the video were **a spatula, a strainer, a** ✓**dessert bowl, cocoa powder, and tissue paper**. the spatula was used to spread the batter in the bowl. the strainer was used to sieve the cocoa powder into the bowl. the dessert bowl was used to hold the batter. the cocoa powder was used to add flavor and color to the batter. the tissue paper was used to wipe the edges of the bowl.

(B) The primary tools and ingredients c used throughout the video were **a mixing bowl, a whisk,** ✗**a baking dish, flour, sugar, eggs, and butte**r. the mixing bowl was used to mix the batter. the whisk was used to beat the eggs. the baking dish was used to bake the cake. the flour was used to add structure to the cake. the sugar was used to add sweetness to the cake. the eggs were used to add moisture to the cake. the butter was used to add flavor and richness to the cake.

(C) The primary tools and ingredients c used throughout the video were **a muffin tin, a spoon, chocolate chips, flour, sugar, eggs, and butter.** the muffin tin was used to bake the muffins. the spoon was used to scoop the batter into the muffin tin. the chocolate chips were used to add flavor and color to the muffins. the flour was used to add structure to the muffins. the sugar was used to add sweetness to the muffins. the eggs were used to add moisture to the muffins. the butter was used to add flavor and richness to the muffins.

(D) The primary tools and ingredients c used throughout the video were **a cake pan, a spatula, chocolate frosting, flour, sugar, eggs, and butter**. the cake pan was used to bake the cake. the spatula was used to spread the frosting on the cake. the chocolate frosting was used to add flavor and color to the cake. the flour was used to add structure to the cake. the sugar was used to add sweetness to the cake. the eggs were used to add moisture to the cake. the butter was used to add flavor and richness to the cake.

(E) The primary tools and ingredients c used throughout the video were **a mixing bowl, a whisk, a baking sheet, flour, sugar, eggs, and butter**. the mixing bowl was used to mix the batter. the whisk was used to beat the eggs. the baking sheet was used to bake the cookies. the flour was used to add structure to the cookies. the sugar was used to add sweetness to the cookies. the eggs were used to add moisture to the cookies. the butter was used to add flavor and richness to the cookies.

Figure 14: Case study on EgoSchema (w/o CoT), demonstrating that Time-R1 achieves better performance than the base model.

## Task

Classify input queries into exactly one of the following categories based on their semantic content:

**1. Human Action (Simple)**

- Definition: Singular physical movements or basic interactions.
- Examples: - person opens a book over their head. - The person gets out some ginger. - who did I talk to in the shopping mall?

**2. Human Action (Complex)**

- Definition: Single continuous event with intricate components or concurrent elements.
- Examples: - He is talking while several people are using rowing machines.
  - One man wearing blue shirt wearing a jumping leg extension and another man wearing red pants play on a field.
  - who did I interact with when I did activity of fixing camping tent?

**3. Human Action (procedural)**

- Definition: contains multiple sequential events with explicit temporal boundaries. contains multiple actions, each with a clear start and end.
- Examples: - The person procures a condiment from the pantry, takes a spoon from the drawer which he uses to scoop it into the pan, then returns the condiment to the pantry, places the spoon in the sink and again stirs the pan.
  - The person takes out a spoon from the drawer, scoops some sugar into the glass, stirs it with the juice, and returns the package to the pantry.
- Negative Examples: - Then the man juices some lemons in a juicer: only one action
  - She gets out a cutting board and knife: only one action
  - He then finishes by doing tricks: only one action
  - She removes bits of shell until there is a small hole: only one action

**4. Human Pose**

- Definition: Static body positions or group configurations. Posture descriptors, positional prepositions
- Examples: - Several other people are in the background working out on the equipment.
  - A young child is seen standing before a set of monkey bars.

**5. Object Existence (Simple)**

- Definition: Current location/status queries. Simple location prepositions.
- Examples: - Where is the tap?
  - where is the chopsticks?
  - In what location did i see the blue tent?

**6. Object Existence (Complex)**

- Definition: Queries about historical object positions changed by human actions, requiring temporal-action context (e.g., "after/before [action]").
- Examples: - Where was the spatula after I first used it?
  - Where was the sieve before I picked it?
  - what bolt did I pick?
  - What mushroom did i chop

**7. Object Attribute**

- Definition: Physical/abstract property inquiries. Property descriptors (color/size/material)
- Examples: - what material did I pick from the shelf?
  - what color is the toilet bin?

**8. Object Counting**

- Definition: Quantitative object presence queries. Numeric quantifiers, plural objects
- Examples: - how many tissue paper were on the floor?
  - how many rolls are in the tray

**9. Object Transition**

- Definition: State/position change confirmation. Transformation verbs, completion checks
- Examples: - The bulb is broken apart.
  - Did I close fridge?,

**10. Environment Change**

- Definition: Dynamic scene modifications. Transient elements, overlay content
- Examples: - video ends with clothes/captions scrolling down

**11. Environment State**

- Definition: Persistent scene elements. Static overlays, permanent fixtures
- Examples: - Intro states 'Progression: Lisa's First Season'
  - 'Trend Routing Technology' logo appears

## Output Format

Return ONLY the exact category name from:[Human Action (Procedural), Human Action (Complex), Human Action (Simple), Human Pose, Object Existence (Simple), Object Existence (Complex),Object Attribute, Object Counting, Object Transition, Environment Change, Environment State]'''

INPUT_PROMPT = '''Given the query below, classify it into one of the categories mentioned above.Query: {query}  Your response:

Figure 15: Prompts for LLM used to annotate the semantics of each query on TVGBench.

