# OpenReview forum: "Time-R1: Post-Training Large Vision Language Model for Temporal Video Grounding"
_NeurIPS.cc/2025/Conference — NeurIPS 2025 poster_

### Official Review · Reviewer_okJ6 · 2025-06-28

**Clarity:** 3
**Significance:** 3
**Originality:** 2
**Rating:** 5
**Confidence:** 4

**Summary:**

This paper applies GRPO (Guided Reinforcement Policy Optimization) to the problem of Temporal Video Grounding, which involves identifying the precise time interval during which a specified event occurs in a video. The authors also make strong contributions on the data side by constructing a benchmark tailored to this task through careful modification of existing datasets. Their approach demonstrates strong performance across multiple downstream tasks.

However, I feel the post-training component is less compelling. The proposed RL method is largely similar to standard GRPO, differing mainly in the introduction of a new reward function (tIOU), which is effectively a form of reward shaping—a fairly common practice in RL. This makes it feel like an application of GRPO to a new task, but with limited technical contribution and novelty.

**Questions:**

- Can the authors ablate the reward function proposed? Here are some simple reward functions that I can think of. In interest of time it may not be easy to complete all, but at least some ablations are important to analyze the impact of the reward function:
   - A simple GRPO style validation, i.e., is the proposed time interval an exact match?
   - Absolute Error or RMSE
   - The second term in the proposed tIoU
- After seeing the use of tIoU vs IoU in the Appendix, there seems to be no clear quantitative reason why the proposed reward function should be used. Are there any qualitative examples?

**Ethical Concerns:**

["NO or VERY MINOR ethics concerns only"]

**Final Justification:**

Overall, I believe this is solid applied research that applies GRPO to a novel problem. While the methodological novelty is limited, the authors and other reviewers have highlighted several other strengths of the paper. Based on the discussion period, I have decided to raise my score to 'Accept'.

**Limitations:**

Yes

**Quality:**

3

**Strengths And Weaknesses:**

**Strengths**

* The paper is clearly written and easy to follow.
* The contributions are broad, covering multiple aspects including dataset construction, benchmarking, and post-training.

**Weaknesses**
Most of my concerns are focused on the novelty and technical contributions of the post-training component:

* The proposed post-training approach using RL is very similar to GRPO, differing primarily in the choice of reward function (the proposed tIOU). This is a fairly minor contribution on its own, since GRPO is designed to allow simple task-specific reward shaping through human-defined or heuristic reward functions.
* Relatedly, no ablation or analysis is demonstrating why the proposed reward function is better than simpler, more intuitive alternatives.

---

> ### Author Rebuttal · Authors · 2025-07-31
>
> We appreciate the time and effort you invested in providing these detailed observations, questions, and comments. We have carefully considered your comments and outlined our responses and proposed revisions below. We hope these clarifications and revisions address your concerns and further enhance the manuscript.
>
> **W1: This work's post-training strategy is almost the same as GRPO.**
>
> - Time-R1 is the first work to introduce the RLVR-style post-training paradigm into temporal video grounding. We address the critical yet underexplored challenge TVG task in this domain by proposing a novel paradigm that fundamentally redefines how long-video understanding can be approached. Our method not only offers a new perspective but also achieves state-of-the-art performance.
> - Our contribution lies in a complete post-training framework that includes Time-R1, TimeRFT, and TVGBench. GRPO serves as a key training objective within Time-R1, but is just one component of the overall comprehensive design.
> - Moreover, we provide detailed ablations to dissect GRPO's design, such as in Appendix Table S3: Ablation of KL and Thinking Process in GRPO; Table S4: Comparison of DAPO and GRPO; and Ablation Study on Reward Design added during rebuttal below.  These insights contribute to a deeper understanding of how RL paradigm impacts long-video understanding reasoning.
>
> **W2 & Q1: Missing ablation to validate the reward design.**
>
> Following your suggestions, we provide a more comprehensive ablation on reward design to demonstrate the effectiveness of our tIoU reward:
>
> - original IoU reward: $r\_{\rm{IoU}}=\frac{[t\_s,t\_e]\cap[t\_s',t\_e']}{[t\_s,t\_e]\cup[t\_s',t\_e']}$
> - only the second term of the tIoU-based reward: $r\_{\rm{tIoUsec}}=1-\frac{|t\_s-t\_s'|}{t}\cdot\frac{|t\_e-t\_e'|}{t}$
> - Exact Matching: $r\_{\rm{em}}=1\ \rm{if\ t\_s=t\_s'\ and\ t\_e=t\_e'\ else\ }0$
> - Absolute Error: $r\_{\rm{abs}}=1 - \frac{|t\_s-t\_s'| + |t\_e-t\_e'|}{2t}$
> - RMSE: $r\_{\rm{rmse}}=1 - \sqrt{(\frac{t\_s-t\_s'}{t})^2 + (\frac{t\_e-t\_e'}{t})^2}$
>
>
> All ablation studies here include the format reward, we omit it for simplicity.
>
> |                                   |        | TVGBench |        |      |        | Charades |        |      |        | ActivityNet |        |      |
> |-----------------------------------|--------|----------|--------|------|--------|----------|--------|------|--------|-------------|--------|------|
> | Model                             | R1@0.3 | R1@0.5   | R1@0.7 | mIoU | R1@0.3 | R1@0.5   | R1@0.7 | mIoU | R1@0.3 | R1@0.5      | R1@0.7 | mIoU |
> | $r\_{\rm{tIoU}}$ (ours) | **41.8**| **29.4**     | **16.4**   | **29.2** | **78.1**   | **60.8**     | **35.3**   | **58.1** | **58.1**  | **39.0**        | **21.4**   | **40.5** |
> | $r\_{\rm{IoU}}$                       |41.4 |28| 15.8| 28.4| 77.6   | 58.9| 32.7| 56.4| 58.8| 38.8| 20.7|39.4|
> | $r\_{\rm{tIoUsec}}$                    | 37.6   | 25.9     | 15.0   | 26.3 | 75.6   | 57.7     | 31.9   | 51.3 | 52.5   | 34.1        | 18.5   | 36.6 |
> | $r\_{\rm{em}}$                         | 26.5   | 16.8     | 9.1    | 18.3 | 61.6   | 40.3     | 18.7   | 40.4 | 35.9   | 22.5        | 13.3   | 26.4 |
> | $r\_{\rm{abs}}$                        | 39.1   | 27.8     | 14.8   | 27.4 | 76.2   | 59.5     | 33.9   | 52.3 | 51.4   | 32.9        | 17.8   | 35.7 |
> | $r\_{\rm{rmse}}$                       | 38.9   | 27.0     | 15.8   | 27.2 | 75.4   | 58.6     | 33.1   | 51.5 | 51.2   | 32.7        | 17.8   | 35.5 |
>
> Experimental results show that our tIoU outperforms all candidate methods.
>
> **Q2: Qualitative examples to illustrate advantage of tIoU over IoU.**
>
> Compared to IoU, tIoU offers a more reasonable definition by introducing a penalty for unreasonable boundary predictions. As shown in our above table in response to Q1, tIoU consistently outperforms IoU across all three benchmarks. Since adding videos or images is not allowed during the rebuttal phase, we will include qualitative examples demonstrating the advantages of tIoU over IoU in the appendix in our revised version.

---

> > ### Comment · Reviewer_okJ6 · 2025-08-04
> > **Response to rebuttal**
> >
> > The authors have addressed my concerns, but I still find the gains over IoU to be negligible, even after reviewing the rebuttal. In my view, using IoU without modification would have made a stronger case for the effectiveness of GRPO-style RLVR, as it leverages a widely accepted metric.
> >
> > That said, the paper has several other strengths, and after considering the perspectives of other reviewers, I have decided to increase my score to accept. I encourage the authors to include ablations and further discussion in the main paper.
> >
> > Thank you.

---

> > > ### Author Response · Authors · 2025-08-04
> > > **Appreciation to Reviewer okJ6 for the Valuable Feedback and Recognition**
> > >
> > > Dear reviewer okJ6,
> > >
> > > We sincerely appreciate your recognition of our efforts to address your concerns. Your constructive and insightful suggestions have been instrumental in improving the quality of our work. We are also grateful for the revised score and thank you for acknowledging the progress we have made.
> > >
> > > As your suggested, we will integrate the new experiments and discussions, such as the choice of reward, from the rebuttal into the main paper to ensure completeness and clarity.
> > >
> > > We welcome any further discussion and would be happy to provide additional clarification if needed.
> > >
> > > Best regards,
> > >
> > > Authors

---

### Official Review · Reviewer_iVA4 · 2025-07-01

**Clarity:** 3
**Significance:** 3
**Originality:** 3
**Rating:** 4
**Confidence:** 3

**Summary:**

This paper introduces Time-R1, a post-training framework based on reinforcement learning (RL) designed to enhance the temporal video grounding (TVG) capabilities of large vision-language models (LVLMs).
To enable efficient RL-based training, the authors propose an appropriate data curation and filtering strategy, as well as a reward model, and further introduce a compact benchmark to validate their approach.
The method achieves state-of-the-art performance across various benchmarks.

**Questions:**

1. Efficiency of Training: SFT vs. Time-R1
How does the computational cost and convergence speed of training a model using large-scale supervised fine-tuning (SFT) compare to training with the curated RL-friendly data in Time-R1? Specifically, is Time-R1 more efficient in terms of GPU hours, memory usage, or time to convergence, given the smaller yet carefully selected dataset?

2. (Minor) In Supplementary Table S5, does “TimeZero-7B” refer to the Time-R1 method with the Qwen2.5-VL-7B backbone, or is it a different model?

**Ethical Concerns:**

["NO or VERY MINOR ethics concerns only"]

**Final Justification:**

The rebuttal has addressed most of my concerns. While I recognize some limitations based on other reviews, I acknowledge the various contributions of the paper and therefore give a borderline accept. I hope the authors incorporate the rebuttal feedback to produce a strong final version.

**Limitations:**

yes

**Quality:**

3

**Strengths And Weaknesses:**

Strengths

1. Time-R1 and TimeRFT

This work pioneers the use of reinforcement learning (RL)-based post-training for large vision-language models (LVLMs) in the context of temporal video grounding (TVG).

2. Data Efficiency and Strong Generalization

The proposed method demonstrates that with a curated, moderately sized RL-friendly dataset (TimeRFT), it is possible to surpass the performance of large-scale supervised fine-tuning approaches.

3. TVG Benchmark
The paper constructs a new, systematic benchmark (TVGBench) specifically designed for temporal video grounding. TVGBench reflects real-world task complexity and includes a balanced distribution of query types (human, object, environment, etc.), supporting objective and reproducible evaluation across diverse model architectures.

Weaknesses

1. Attribution of SOTA Performance to Backbone Quality
It remains unclear whether the observed SOTA performance is primarily due to the use of a stronger LVLM backbone (Qwen2.5-VL-7B), rather than the proposed training and reward strategies. In contrast, the SFT baselines utilize a variety of vision encoders and language models (e.g., TimeChat: ViT-G/14 from EVA-CLIP + LLaMA-2-7B; TIMESUITE: UMT-L + Mistral-7B; VideoChat-Flash: UMT-L + Qwen2-7B; TRACE: ViT-L/14 + Mistral-7B). The main results do not directly demonstrate whether Time-R1 yields significant performance improvements over these baselines when using the same vision encoder and language model architecture. A controlled head-to-head comparison on identical backbones would be necessary to fairly isolate the effectiveness of the proposed method.

2. Effectiveness of the tIoU Reward
Section 3.2 introduces the tIoU reward to promote deeper temporal understanding. However, in the ablation study (Table 2, Row 1 vs. Row 2), adding tIoU appears to decrease performance. The authors should provide a more detailed analysis or hypothesis as to why incorporating tIoU does not consistently yield gains, and under what circumstances (e.g., data distribution, filtering, w/ Multi-Epoch) it becomes effective.

3. Ablation of Reasoning Template Reward
While the paper claims that the reasoning template reward encourages better intermediate reasoning and thus benefits temporal grounding, it is not empirically clear to what extent this mechanism enhances the model's reasoning capability or TVG performance. A direct ablation comparing models with and without the reasoning template reward on reasoning quality, as well as an assessment of its impact across various benchmarks (as in Table 1 and Table S1), would greatly strengthen the claims.

4. Quantitative Comparison of Inference Speed
The supplementary material mentions the slow inference speed as a limitation. However, a quantitative comparison of inference time or throughput between Time-R1 and relevant baselines is missing. Such a comparison would help practitioners assess the real-world trade-offs between accuracy and efficiency.

5. (minor) typo in Table 2, trainning -> training

---

> ### Author Rebuttal · Authors · 2025-07-31
>
> We appreciate the time and effort you invested in providing these detailed observations, questions, and comments. We have carefully considered your comments and outlined our responses and proposed revisions below. We hope these clarifications and revisions address your concerns and further enhance the manuscript.
>
> **W1: Could the SOTA performance be attributed to a stronger backbone rather than the proposed training method and reward design?**
>
> In the following experiments, we aim to show that (1) open-sourced academic models, like TimeSuite, can benefit from our RL post-training; and (2) many well-pretrained model across model sizes and backbones can be substantially improved via our training procedure.
>
> We conducted additional experiments with RL-based post-training on other base models, including **TimeSuite-7B, Qwen-2.5-VL-3B, MIMO-VL-7B, InternVL-2B, and InternVL-8B**.  We post-train all base models using our 2.5K data filtered by Qwen-2.5-VL-7B. The results are presented as follows:
>
> |||| TVGBench  |||| Charades  |||| ActivityNet |||
> |-|-|-|-|-|-|-|-|-|-|-|-|-|-|
> | Model | Size | R1@0.3| R1@0.5| R1@0.7| mIoU | R1@0.3| R1@0.5| R1@0.7 | mIoU| R1@0.3| R1@0.5 | R1@0.7| mIoU  |
> | Qwen-2.5-VL-3B (Base) | 3B | 11.5 | 6.5  | 3.8  | 8.3  | 24.2  | 15.5 | 8.1| 16.3| 13.0 | 7.1| 3.3| 9.8  |
> | Qwen-2.5-VL-3B (Time-R1) | 3B| 33.5 | 21.0  | 10.5 | 21.7 | 74.6 | 53.1  | 26.0 | 51.2 | 40.0 | 21.0  | 8.7| 23.2|
> | Δ ↑  |  | **+22.0** | **+14.5** | **+6.7** | **+13.4** | **+50.4** | **+37.6** | **+17.9** | **+34.9** | **+27.0** | **+13.9**| **+5.4** | **+13.4** |
> | Qwen-2.5-VL-7B (Base)    | 7B| 24.9  | 16.0  | 8.0  | 16.3  | 58.7 | 38.3 | 16.6 | 37.9      | 34.3 | 21.6  | 12.9  | 22.9 |
> | Qwen-2.5-VL-7B (Time-R1) | 7B   | 41.6| 28.5 | 15.6| 28.6  | 77.4 | 60.0   | 34.1 | 57.2  | 56.2  | 37.4 | 20.4 | 38.0 |
> | Δ ↑ |  | **+16.7** | **+12.5** | **+7.6** | **+12.3** | **+18.7** | **+21.7** | **+17.5** | **+19.3** | **+21.9** | **+15.8**| **+7.5** | **+15.1** |
> | MiMo-VL-7B (Base) | 7B   | 22.4 | 12.6| 6.6  | 15.7  | 48.5 | 27.0 | 12.1  | 31.7 | 31.3  | 19.7  | 12.1  | 24.2 |
> | MiMo-VL-7B (Time-R1)  | 7B   | 41.2   | 27.8 | 15.1| 27.4| 79.9| 63.9| 33.4 | 53.9  | 45.6| 27.2  | 14.2 | 31.9 |
> | Δ ↑  || **+18.8** | **+15.2** | **+8.5** | **+11.7** | **+31.4** | **+36.9** | **+21.3** | **+22.2** | **+14.3** | **+7.5**    | **+2.1** | **+7.7**  |
> | InternVL-2B (Base)  | 2B | 16.3 | 6.3| 2.3 | 11.7  | 20.9   | 7.8  | 1.9  | 15.4 | 18.6  | 8.5   | 3.1 | 14.2 |
> | InternVL-2B (Time-R1)    | 2B   | 21.8 | 9.5  | 4.1 | 14.8 | 24.0 | 11.5 | 3.5  | 15.7 | 20.6  | 9.5  | 3.9  | 14.2 |
> | Δ ↑ | | **+5.5**  | **+3.2**  | **+1.8** | **+3.1**  | **+3.1**  | **+3.7**  | **+1.6**  | **+0.3**  | **+2.0**  | **+1.0**| **+0.8** | **+0.0**  |
> | InternVL-8B (Base) | 8B| 17.4  | 8.3  | 3.4 | 11.8 | 27.8| 11.9 | 3.7 | 20.6  | 33.1| 18.4| 10.3| 24.0 |
> | InternVL-8B (Time-R1)| 8B| 38.0 | 22.5 | 9.2  | 24.2 | 70.0| 45.1 | 18.3| 44.1| 46.8 | 25.9 | 11.7  | 31.1  |
> | Δ ↑  | | **+20.6** | **+14.2** | **+5.8** | **+12.4** | **+42.2** | **+33.2** | **+14.6** | **+23.5** | **+13.7** | **+7.5**| **+1.4** | **+7.1**  |
> | TimeSuite-7B (Base) | 8B| 31.1  | 18.0| 8.8| 21.6 | 70.1 | 48.5|24.2| 45.7|-|-|-|-|
> | TimeSuite-7B (Time-R1)| 8B| 33.3 | 19.7 | 9.2 | 22.3 | 71.7 | 49.5 | 24.2| 47.2|-|-|-|-|
> | Δ ↑  | | **+2.2** | **+1.7** | **+0.4** | **+0.7** | **+1.6** | **+1.0** | **+0.0** | **+1.5** |-|-|-|-|
>
> We observe several key findings, revealing that our approach is generally applied to many base models.
>
> First, **models of different sizes can also benefit from our approach**. For instance, the Qwen-VL-3B model achieves a +13.4 mIoU improvement on TVGBench and a +34.9 mIoU improvement on Charades. This is attributed to the fact that our training data is distilled based on filtering from Qwen-VL-7B, demonstrating that our RL-based method can effectively learn from the data.
>
> Second, **models with different backbones also benefit**. For example, InternVL-8B achieves a +12.4 mIoU on TVGBench and +23.5 on Charades, while MiMo-VL-8B improves by +11.7 and +22.2 on TVGBench and Charades, respectively.
>
> Additionally, we notice that **when using weaker model with a different backbone, the transferability of the training data becomes less effective, but still beneficial**. For example, training InternVL-2B on data filtered by Qwen-2.5-VL-7B only yields marginal gains, improving only +3.1 on TVGBench and +0.3 on Charades. Similar phenomenon is observed on TimeSuite-7B backbone. This is mainly due to architectural differences, base model's intelligence, and the data filtered by Qwen-2.5-VL-7B are not well-suited for RL training on InternVL-2B. However, we expect that filtering the data using base model itself could yield improvements comparable to those observed when transferring from Qwen-2.5-VL-7B to Qwen-2.5-VL-3B.
>
> **W2: What is the role of tIoU and under what conditions is it most effective?**
>
> **tIoU is particularly effective when combined with multi-epoch (ME) training. Further including Gaussian filtering and sample filtering consistently contribute to the performance gains.**
>
> Comparing Row 2 and Row 6 in Table 2, the combination of tIoU and ME achieves the best performance on R1@0.7, with a score of 16.4. In contrast, using IoU with ME (Row 4) yields lower performance on R1@0.7 of 14.2.
>
> In Appendix Figure S2, tIoU consistently improves performance as the number of training epochs increases, and tIoU outperforms IoU in the multi-epoch training. We attribute this to the superior boundary sensitivity of tIoU.
>
> In addition, Gaussian filtering (GF) consistently improves model performance, and its effectiveness is further amplified when combined with tIoU. Specifically, comparing Row 1 and Row 3 (IoU + GF), as well as Row 2 and Row 6 (tIoU + GF), we observe that tIoU leads to a performance gain of +3.5%, whereas IoU yields only a marginal improvement of +0.2%. Furthermore, sample filtering, as shown in the last row, provides additional stability to the model’s performance, demonstrating its complementary benefit in our training pipeline.
>
> **W3: The paper lacks ablation studies on the template reward function.**
>
> Below, we provide an ablation of the format reward.
> Comparing row 1 and row 2, the format reward consistently improves the performance. While training without reasoning format reward would lead to better performance on R1@0.3/0.5/0.7 on Charades, the actual mIoU of training with format reward outperforms training without format reward.
>
> |||TVGBench||||Charades||||ActivityNet|||
> |-|-|-|-|-|-|-|-|-|-|-|-|-|
> | Reward | R1@0.3 | R1@0.5| R1@0.7 | mIoU | R1@0.3 | R1@0.5| R1@0.7 | mIoU | R1@0.3 | R1@0.5| R1@0.7 | mIoU |
> | $r\_{\rm{tIoU}}+r\_{\rm{format}}$ (ours) | **41.8**| **29.4**| **16.4**| **29.2** | 78.1| 60.8| 35.3| **58.1** | **58.1**  | **39.0**| **21.4**   | **40.5** |
> | $r\_{\rm{tIoU}}$| 40.5| 27.6| 15.4| 28.0 | **78.4**   | **61.6**| **36.1**| 54.0 | 55.2| 36.8| 20.2   | 38.4 |
> | $r\_{\rm{format}}$| 27.1| 18.0| 10.1| 18.8 | 60.4| 39.2| 17.4   | 39.3 | 35.1| 22.2| 13.1   | 26.0 |
>
> Moreover, if you have interest, we provide a more comprehensive ablation study on the choice of tIoU in response to Reviewer okJ6.
>
> **W4: The inference speed of this method seems slow. Is there any comparison available?**
>
> To address this, we conduct a comparison of inference speeds with and without CoT reasoning process. We adapt our model for implementing both vLLM and HuggingFace Transformers library, and evaluate the inference speed under both frameworks. The results below demonstrate that with vLLM library, our method achieves substantial improvements in inference efficiency, significantly reducing latency while maintaining strong performance.
>
> | test with/without CoT using 8×H100 | TVGBench| Charades| ActivityNet|
> |-|-|-|-|
> | Sample Num| 800| 3720| 17031|
> | vLLM inference time| 8.3min/6.9min | 11.7min/11.2min | 36.1min/33.2min |
> | transformers inference time| 42min/27.3min | 3.3hour/1.2hour | 15hour/7.5hour  |
>
> Therefore, **for a 1.5-minute input video sampled at 2 fps, our model—when deployed with vLLM—achieves an average inference time of 4.98 seconds with CoT and 4.14 seconds without CoT**, which is actually not too slow.
> Given that most models adopt similar MLLM architectures, their inference speed is comparable with ours inference without CoT.
>
> **Q1: Comparison of training efficiency between RL and SFT.**
>
> We compared RL and SFT across various efficiency metrics, including training time, memory usage and experimental results below.
> To further demonstrate the data efficiency of the RL paradigm, we trained an SFT model with full size 339K TVG samples and found that it still underperforms compared to our RL model trained with only 2.5K samples. The SFT-LoRA-2.5K only underperforms SFT-LoRA-339K slightly, and even slightly surpasses SFT-LoRA-339K on ActivityNet.
>
> | Model  | TrainTime | MaxGPUMem | | TVGBench ||||Charades||||ActivityNet|||
> |-|-|-|-|-|-|-|-|-|-|--|-|-|-|-|
> |||| R1@0.3 | R1@0.5   | R1@0.7 | mIoU | R1@0.3 | R1@0.5| R1@0.7 | mIoU | R1@0.3 | R1@0.5| R1@0.7 | mIoU |
> | RL-2.5K (5 epochs)| 15h|78G| **41.8**| **29.4**| **16.4**| **29.2**| **78.1**| **60.8**| **35.3**| **58.1**| **58.1**| **39.0**| **21.4**| **40.5** |
> | SFT-LoRA-2.5K (5 epochs) | 3.26h| 76G| 38.6| 24.5| 14.6| 25.9 | 74| 55.7| 29.9   | 53.2 | 52.3| 34.3| 18.8   | 35.1 |
> | SFT-LoRA-339K (1 epoch)  | 5d2h| 76G| 38.9| 28.2| 15.2| 27.4 | 72.4| 54.7| 26.9   | 51.7 | 45.7   | 29.3| 16.4   | 30.5 |
>
> These results highlight: (1) the strong potential of RL to achieve higher performance with significantly less data. (2) our data filtering strategy is both effective for RL and SFT. While RL incurs higher computational cost, its superior performance with minimal data suggests strong potential for future advancements.
>
> **Q2: Typo in Table 2 and in Supplementary Table S5**
>
> We apologize that "TimeZero-7B" is a typo. It should refer to the "Time-R1-7B". We will fix these typos in our revised version.

---

> ### Comment · Reviewer_iVA4 · 2025-08-04
> **Response to rebuttal**
>
> I appreciate the authors’ kind and detailed responses, which have addressed most of my concerns.
>
> While considering other reviews as well, I would like to point out as a weakness that the performance improvement from the proposed tIoU is not marginal, and that too many techniques (tIoU, format, Gaussian filtering, multi-epoch, sample filtering, etc.) are combined empirically.
>
> I have decided to increase my score to borderline accept. I hope that by reflecting on the rebuttal well, the paper will be finalized as a strong work.

---

> > ### Author Response · Authors · 2025-08-04
> > **Appreciation to Reviewer iVA4 for the Valuable Feedback and Recognition**
> >
> > Dear reviewer iVA4,
> >
> > We sincerely appreciate your recognition of our efforts to address your concerns. Your constructive and insightful suggestions have been instrumental in improving the quality of our work. We are also grateful for the revised score and thank you for acknowledging our progress.
> >
> > Regarding the weakness of our used techniques, we aim to achieve state-of-the-art performance to enhance the impact of the RL paradigm. We will discuss potential technical limitations in the Discussion section.
> >
> > We welcome any further discussion and are glad to provide additional clarification if needed.
> >
> > Best regards,
> >
> > Authors

---

### Official Review · Reviewer_1QoR · 2025-07-02

**Clarity:** 3
**Significance:** 3
**Originality:** 3
**Rating:** 5
**Confidence:** 5

**Summary:**

This paper tackles the problem of Temporal Video Grounding (TVG) with Large Vision-Language Models (LVLMs). The authors claim that existing LVLM-based TVG methods are sub-optimal (sometimes even perform worse than conventional feature-based small models) due to the over-penalization of false negatives during supervised fine-tuning (SFT). Therefore, to effectively supervise the model through task-specific metrics (e.g., temporal IoU), the authors proposed to leverage RL to allow direct optimization, thereby reducing rigid penalties of autoregressive losses and encouraging plausible timestamp predictions. The contributions of this work include Time-R1, TimeRFT, and TVGBench.

**Questions:**

Please refer to the weakness section for my questions. My major concern is the lack of clarifications and important baselines.

**Ethical Concerns:**

["NO or VERY MINOR ethics concerns only"]

**Final Justification:**

After reading the rebuttal and discussing with the authors, my concerns about this paper have been resolved. Given the good quality and detailed experiments of this paper, I would recommend acceptance of the paper and encourage the authors to include these discussions in their final version.

**Limitations:**

The limitations are addressed in the supplementary material, but the potential negative societal impacts are not.

**Quality:**

3

**Strengths And Weaknesses:**

Strengths:

1. Overall, the paper is well-written and easy to follow.
2. The motivation for using RL is clear and reasonable -- SFT overly penalizes false negatives even when the predicted timestamps are close to the ground truth, and adopting additional timestamp prediction heads would sacrifice the pretrained language capabilities of LLMs.
3. I like the idea of dynamic hard sampling strategy during training. The experimental results also demonstrated the effectiveness of this scheme.

Weaknesses:

1. It is unclear why applying GRPO (DAPO) with temporal grounding data & tIoU rewards can improve general video QA performance. To my understanding, training solely on such data would improve performance on TVG, but would also lead to lower performance on other tasks. The authors should provide more clarification for it.
2. In Figure 4, why does SFT significantly underperform even the base model on all tasks?
3. For the SFT-based cold start, the authors did not provide an in-depth analysis on why using dense video captioning-style thinking as CoT.
4. Several recent important baselines / benchmarks shall be carefully discussed / compared to justify both the effectiveness of Time-R1 and TVGBench. To name some, the missing methods in Table 1 and related work are [1-5], the similar benchmarks requiring further discussions are [2, 7, 8].

[1] Momentor: Advancing Video Large Language Model with Fine-Grained Temporal Reasoning. ICML 2024.
[2] E.T. Bench: Towards Open-Ended Event-Level Video-Language Understanding. NeurIPS 2024.
[3] Task Preference Optimization: Improving Multimodal Large Language Models with Vision Task Alignment. CVPR 2025.
[4] VTG-LLM: Integrating Timestamp Knowledge into Video LLMs for Enhanced Video Temporal Grounding. AAAI 2025.
[5] VideoMind: A Chain-of-LoRA Agent for Long Video Reasoning. arXiv 2025.
[6] VideoExpert: Augmented LLM for Temporal-Sensitive Video Understanding. arXiv 2025.
[7] ReXTime: A Benchmark Suite for Reasoning-Across-Time in Videos. NeurIPS 2024.
[8] CG-Bench: Clue-grounded Question Answering Benchmark for Long Video Understanding. ICLR 2025.

---

> ### Author Rebuttal · Authors · 2025-07-31
>
> We appreciate the time and effort you invested in providing these detailed observations, questions, and comments. We have carefully considered your comments and outlined our responses and proposed revisions below. We hope these clarifications and revisions address your concerns and further enhance the manuscript.
>
> **W1: Why does the TVG task help improve VQA performance?**
>
> **From the data perspective**, the TVG task equips the model with the ability to localize relevant segments based on a textual query. This capability is crucial for long-video VQA, as it helps the model focus on the most pertinent moments and reduces the distraction from irrelevant or redundant information.
>
> **From the learning perspective**, our RL-based training mitigates the risk of overfitting to noisy or false negative data by allowing the model to learn from its own generated samples, enhancing the generalization ability of the model. It is the RL loss design and training paradigm that incentivizes model's capability without sacrificing its original ability.
>
> Furthermore, prior work such as TimeSuite[ICLR'25] has shown that jointly training on TVG + VQA data via supervised fine-tuning (SFT) significantly boosts VQA performance. This indirectly validates the utility of the TVG task. Our RL-based paradigm achieves similar benefits without relying on any VQA data, highlighting its ability to enhance general-purpose video understanding.
>
> **W2: Why does SFT underperforms base model?**
>
> We have provided some clarifications and additional results in Appendix Figure S1. Specifically, we implemented two versions of supervised fine-tuning: Full-parameter fine-tuning of the LLM (denoted as SFT) and LoRA-based fine-tuning of the LLM (denoted as SFT-LoRA). Our RL training full-finetunes the LLM.
> We present the detailed experiment results below:
>
> | Model    |        | TVGBench |        |      |        | Charades |        |      |        | ActivityNet |        |      |
> |----------|--------|----------|--------|------|--------|----------|--------|------|--------|-------------|--------|------|
> |          | R1@0.3 | R1@0.5   | R1@0.7 | mIoU | R1@0.3 | R1@0.5   | R1@0.7 | mIoU | R1@0.3 | R1@0.5      | R1@0.7 | mIoU |
> | Base     | 24.9   | 16       | 8      | 16.3 | 58.7   | 38.3     | 16.6   | 37.9 | 34.3   | 21.6        | 12.9   | 22.9 |
> | SFT      | 25.9   | 14.4     | 5.8    | 15.4 | 49.5   | 31.3     | 12.7   | 31.2 | 27.7   | 13.6        | 5.8    | 15.7 |
> | SFT-LoRA | 38.6   | 24.5     | 14.6   | 25.9 | 74     | 55.7     | 29.9   | 53.2 | 52.3   | 34.3        | 18.8   | 35.1 |
> | RL       | **41.8**   | **29.4**     | **16.4**   | **29.2** | **78.1**   | **60.8**     | **35.3**   | **58.1** | **58.1**   | **39.0**        | **21.4**   | **40.5** |
>
> We observed that full fine-tuning via SFT leads to severe overfitting. **While SFT-LoRA partially mitigates this issue and improves performance on TVG tasks, it still leads to a slight degradation in VQA performance as shown in Figure S1**.
>
> In contrast, **RL consistently preserves generalization and improves performance**. For example, on ActivityNet, RL boosts mIoU from 16.3 to 29.2, whereas SFT drops it to 15.4, and SFT-LoRA only improves it to 25.9. On the VideoMME VQA benchmark, RL improves performance from 53.0 to 54.2, while SFT-LoRA decreases it to 51.7.
>
> **W3: Why use dense video captions as cold-start data?**
>
> We observed that, without any cold-start initialization, the model gradually learns to generate video descriptions through Chain-of-Thought (CoT) reasoning. Building on this empirical observation, we employ SFT-based cold-start using dense video captions to accelerate convergence toward the format reward and to reduce the length of the reasoning chains.
>
> **W4: The paper lacks baselines and benchmark comparisons.**
>
> Thanks for your suggestions, we will expand our related work section to include discussion and citation of the suggested methods.
> In addtion, we now provide comparisions against additional baseline methods with corresponding evaluation metrics on Charades and TVGBench. As shown below, our Time-R1 still achieves state-of-the-art results:
> |                          |      |           | Charades|          |           |           ActivityNet |          |           ||
> |--------------------------|------|-------------------|--------|--------|------|---------------|--------|--------|------|
> | Model                     | Size | R1@0.3 | R1@0.5 | R1@0.7 | mIoU | R1@0.3 | R1@0.5 | R1@0.7 | mIoU |
> | **ZeroShot**             |      |                   |        |        |      |               |        |        |      |
> | Momentor                 | 7B    | 42.6              | 26.6   | 11.6   | 28.5 | 42.9          | 23.0   | 12.4   | 29.3 |
> | VideoChat-TPO | 7B    | 58.3              | 40.2   | 18.4   | 38.1 | -             | -      | -      | -    |
> | VTG-LLM                  | 7B    | -                 | 33.8   | 15.7   | -    | -             | -      | -      | -    |
> | VideoMind-2B             | 2B   | 67.6              | 51.1   | 26.0   | 45.2 | 44.0          | 26.5   | 12.6   | 30.1 |
> | VideoMind-7B             | 7B   | 73.5              | 59.1   | 31.2   | 50.2 | 48.4          | 30.3   | 15.7   | 33.3 |
> | VideoExpert | 7B    | 61.5              | 40.3   | 20.9   | 41.1 | -             | -      | -      | -    |
> | Time-R1-3B | 3B   | 74.6              | 53.1   | 26.0   | 51.2 | 40.0          | 21.0   | 8.7    | 23.2 |
> | Time-R1-7B | 7B   | **78.1**              | **60.8**   | **35.3**   | **58.1** | **58.1**          | **39.0**   | **21.4**   | **40.5** |
> | **FineTune**             |      |                   |        |        |      |               |        |        |      |
> | VideoChat-TPO| 7B    | 77.0              | 65.0   | 40.7   | 55.0 | -             | -      | -      | -    |
> | VideoExpert | 7B    | 74.3              | 60.8   | 36.5   | 52.2 | -             | -      | -      | -    |
> | Time-R1-3B | 3B   | 78.7              | 64.1   | 36.9   | 59.9 | 66.8          | 46.8   | 24.7   | 46.1 |
> | Time-R1-7B | 7B   | **82.8**              | **72.2**   | **50.1**   | **60.9** | **73.3**          | **55.6**   | **34.0**   | **52.1** |
>
>
> **Regarding the benchmark comparison**:
> - CG-Bench[8] is a multiple-choice benchmark, which by nature is not a timestamp prediction task.
> - E.T.Bench[2], as shown in Table 4 on Page 11 of its paper, uses TVG data limited to Charades, EgoNLQ, and QVHighlights.
> - RexTime[7] only includes ActivityNet and QVHighlights.
>
> In contrast, our benchmark covers Charades, ActivityNet, EgoNLQ, TaCoS, and HiREST, with balanced distribution on video length, video domain and query center, making it potentially a more comprehensive benchmark for the true TVG task. Besides, we re-annotated each sample with consistent semantics and re-balanced the distribution. This ensures the benchmark's reliability and representativeness, allowing it to serve as a useful evaluation suite.

---

> > ### Comment · Reviewer_1QoR · 2025-08-01
> >
> > Thanks for the response from the authors. Some of my concerns have been resolved, while I still have two questions regarding W2 and W3.
> >
> > W2: What are the hyperparameters you used for SFT(-LoRA) experiments? Did you try to tune the hyperparams to avoid overfitting? Are the current results in the table the optimal setting for SFT?
> >
> > W3: I understand that cold-start initialization is needed for RL. But my concern was why adopt dense video captioning data (rather than other data) for SFT? To my understanding, the questions and responses for dense video captioning are not strictly aligned with the actual use cases for TVG. Are there any better choices regarding the data for cold-start initialization?

---

> ### Author Response · Authors · 2025-08-02
> **Official Comment to Reviewer 1QoR's Further Concerns**
>
> Thank you for your responses. We address your additional comments as follows:
>
> **W2: Hyperparameters for SFT-LoRA and parameter tuning.**
>
> **Our current hyperparameters:**
> For SFT-LoRA, the LoRA rank is set to 64, and the LoRA alpha is set to 128. Both SFT and SFT-LoRA use the following settings: a learning rate of 2e-5, the AdamW optimizer with $\beta\_1$=0.9, $\beta\_2$ = 0.999, a weight decay of 0, the batch size of 8, and accumulation steps of 2. We fine-tune for 5  epochs on our 2.5K data, and use a linear scheduler to gradually decay the learning rate to 0.
>
> **Parameter tuning for SFT-LoRA:**
> To further address your concerns, we provide an ablation study of SFT-LoRA below on the key LoRA hyperparameters, as well as the number of training epochs. We conclude that **under all current configurations, RL training outperform all SFT trainnig settings by a large margin.**
>
> |Ablation of LoRA-rank/alpha|| TVGBench |||| Charades |||| ActivityNet |||
> |-|-|-|-|-|-|-|-|-|-|-|-|-|
> | LoRA rank/alpha| R1@0.3| R1@0.5| R1@0.7| mIoU| R1@0.3   | R1@0.5   | R1@0.7| mIoU| R1@0.3| R1@0.5| R1@0.7| mIoU|
> | Time-R1-7B|**41.6**| **28.5** | **15.6** | **28.6** | **77.4** | **60.0** |**34.1**|**57.2** |**56.2**|**37.4**| **20.4** | **38.0** |
> | 16/32| 31.4| 22.0| 11.5| 21.8| 58.7| 44.9| 25.6| 40.4| -| -| -| -|
> | 32/64| 35.0| 24.2| 14.4| 24.8| 72.4| 54.0| 29.4| 48.8| -| -| -| -|
> | 64/128 (paper reported) | 38.5| 24.5| 14.6| 26.2| 74.0| 55.7| 29.9| 49.9| 52.3| 34.3| 18.8     | 36.5|
> | 128/256| 38.4| 25.4| 14.2| 26.4| 76.5| 58.3| 32.9| 52.2| -| -| -| -|
> | 256/512| 39.1| 27.1| 15.1| 26.9| 75.9| 57.4| 32.7| 51.6| -| -| -| -|
>
> |Ablation of Training Epochs for SFT-LoRA||| TVGBench |||| Charades |||| ActivityNet |        ||
> |-|-|-|-|-|-|-|-|-|-|-|-|-|-|
> || Epochs | R1@0.3 | R1@0.5   | R1@0.7 | mIoU | R1@0.3 | R1@0.5| R1@0.7 | mIoU | R1@0.3 | R1@0.5| R1@0.7 | mIoU |
> || 1      | 38.0   | 25.1| 14.1   | 26.6 | 73.5| 57.3| 33.3| 50.5 | 54.8| 37.1| 21.7   | 39.0 |
> | LoRA rank/alpha=64/128  | 2| 36.9| 24.6| 14.1| 25.9 | 73.5   | 56.2| 31.1   | 50.3 | 52.4   | 35.7| 20.5   | 37.4 |
> || 5| 38.5   | 24.5| 14.6   | 26.2 | 74.0| 55.7| 29.9| 49.9 | 52.3| 34.3| 18.8   | 36.5 |
> | | | | | | | | | | | | | | |
> || 1| 27.4| 18.1| 8.9| 18.3 | 47.8| 33.4| 16.7   | 31.7 | -| -| -| -|
> | LoRA rank/alpha=128/256 | 2| 37.9| 26.2| 14.4 |26.2|76.4| 57.0| 30.7   | 51.3 | -| -| -| -|
> ||5|38.4|25.4|14.2| 26.4 | 76.5| 58.3| 32.9| 52.2 | -| -| -| -    |
>
> Due to time constraints, we are unable to evaluate on ActivityNet. which requires too much time for evaluation.
>
> Regarding the ablation on SFT-LoRA: Our ablation study demonstrates that the SFT-LoRA configurations of 128/256 and 256/512 yield the best performance. Specifically, the 128/256 setting performs better on TVGBench, while 256/512 achieves superior results on Charades. Second, increasing the number of training epochs does not necessarily degrade performance on the TVG task. For instance, the 128/256 setting continues to improve with longer training, whereas the 64/128 setting tends to overfit as training progresses.
>
>
> **W3: The effect of using dense video caption data for SFT cold-start.**
>
> We summarize the role of SFT as a cold-start as follows:
>
> - **Caption data as Chain-of-Thought (CoT) provides video content priors as text:** Intuitively, CoT-style video captions help the model better understand the visual scene and offer a prior over the visual content, which may be beneficial for generating timestamp predictions conditioned on the query. This textual prior makes it easier for the MLLM to infer the query-relevant temporal segments.
> - **Functional role in controlling CoT length:** One key motivation for using SFT is to control the length of the CoT. We empirically observed that the model tends to generate CoTs resembling video captions, so we naturally opted to use caption data to regulate the CoT length.
>
> We present the results of the 3B model here to illustrate the role of SFT as follows:
>
> | | | TVGBench |||| Charades |||| ActivityNet |||
> |-|-|-|--|--|-|--|-|-|-|--|--|--|
> | Model| R1@0.3 | R1@0.5| R1@0.7 | mIoU | R1@0.3 | R1@0.5   | R1@0.7 | mIoU | R1@0.3 | R1@0.5| R1@0.7 | mIoU |
> | w/o Cold-Start | 29| 17.1| 8| 18| 67| 44.9| 21.5| 44.4 | 36.6| 18.8| 7.8| 21.1 |
> | w/ Cold-Start  | 33.5| 21.0| 10.5| 21.7 | 74.6| 53.1| 26.0   | 51.2 | 40.0   | 21.0| 8.7    | 23.2 |
>
> **We leave video CoT and complex reasoning as Future work**: In this work, our focus is introducing a new RL-based paradigm for the TVG task and demonstrating that CoT plays a supportive role. The CoT reasoning is an important direction for video reasoning, and cold-starting with SFT is a critical mechanism for enabling the model to effectively generate more complex CoTs. A promising future direction is to develop models capable of generating more complex CoTs to tackle harder reasoning tasks, such as multi-hop query understanding. We leave the exploration of more effective CoT designs and better cold-start strategies for future work.

---

> > ### Comment · Reviewer_1QoR · 2025-08-03
> >
> > Thanks for the responses from the authors. I have no more questions and will raise the rating to "accept".

---

> > > ### Author Response · Authors · 2025-08-04
> > > **Appreciation to Reviewer 1QoR for the Valuable Feedback and Recognition**
> > >
> > > Dear reviewer 1QoR,
> > >
> > > We sincerely appreciate your recognition of our efforts to address your concerns. Your constructive and insightful suggestions have been instrumental in improving the quality of our work. We are also grateful for the revised score and thank you for acknowledging our progress.
> > >
> > > We welcome any further discussion and are glad to provide additional clarification if needed.
> > >
> > > Best regards,
> > >
> > > Authors

---

### Official Review · Reviewer_XTMy · 2025-07-02

**Clarity:** 3
**Significance:** 2
**Originality:** 2
**Rating:** 4
**Confidence:** 2

**Summary:**

The work introduces RL post-training framework into Temporal Video
Grounding and achieves SOTA performance across various TVG benchmarks.
In addition, the work also curates a dataset and benchmark for TVG task,
contributing to the development of the research community.

**Questions:**

1. The paper only uses Qwen2.5-VL-7B as the base model. Can the same conclusion be reached by using other series models or models of different sizes?
2. Figure 4 shows that the VQA capability is improved after RL post training. Is this because the dataset used in the TimeRFT stage contains VQA data? If only TVG CoT data is used for training in the post training stage, can the VQA capability be kept on par with the base model?
3. The paper does not conduct ablation experiments on the two rewards of the TVG task. Is it possible to use only one of the rewards to observe the experimental results?

**Ethical Concerns:**

["NO or VERY MINOR ethics concerns only"]

**Final Justification:**

Most of my concerns have been addressed. I recommend the authors include the relevant experiments in the revised version. I maintain my current vote.

**Limitations:**

yes

**Paper Formatting Concerns:**

None.

**Quality:**

3

**Strengths And Weaknesses:**

Strengths：
1. This work achieves SOTA performance across TVG benchmarks
2. This work effectively integrates RL into TVG task and innovatively proposes two rewards curated for TVG task to assist the post training

Weaknesses：
1. TVGBench has only 800 samples and comes from existing datasets, so the
necessity of building this benchmark is questionable
2. This work is solid, but it is just the RL post-train paradigm commonly used in LLM
adapted to the TVG task, which is novel but not much

---

> ### Author Rebuttal · Authors · 2025-07-31
>
> We appreciate the time and effort you invested in providing these detailed observations, questions, and comments. We have carefully considered your comments and outlined our responses and proposed revisions below. We hope these clarifications and revisions address your concerns and further enhance the manuscript.
>
> **W1: The necessity of designing TVGBench**
>
> As stated in Figure 3 and Section 3.4 of our paper, existing benchmarks are often either excessively large or overly domain-specific, which poses challenges for effectively evaluating large multimodal models. In contrast, TVGBench is deliberately designed to be compact yet comprehensive. To show its advantage, we illustrate the comparison between benchmarks used in this work:
>
> | test with/without CoT using 8×H100 | TVGBench| Charades| ActivityNet|
> |-----------------------------------------|---------------|-----------------|-----------------|
> | Domain| Mixed| Indoor| HumanActivity|
> | Sample Num| 800| 3720| 17031|
> | vLLM inference speed| 8.3min/6.9min | 11.7min/11.2min | 36.1min/33.2min |
> | transformers inference speed| 42min/27.3min | 3.3hour/1.2hour | 15hour/7.5hour  |
>
> Noting that TVGBench encompasses a broader range of data sources, we built our implementation on top of the vLLM library to achieve faster inference speed. As our framework would be fully open-source, it provides a useful suite for the community.
>
> Moreover, although TVGBench is constructed by curating samples from existing datasets, we re-annotated each sample with consistent semantics and re-balanced the distribution. This ensures the benchmark's reliability and representativeness, allowing it to serve as a useful evaluation suite.
>
> **W2: Solid work, novel but not much**
>
> Thank you for recognizing the solidity and novelty of our work. We address a crucial and underexplored challenge in long-video understanding, proposing a new RL paradigm that fundamentally shifts how this problem is approached. Our method not only introduces a fresh perspective but also achieves state-of-the-art performance, demonstrating both its practical effectiveness and scientific contribution. Furthermore, we will publicly release all our training data, training code, and inference code, which we believe will be valuable to the research community.
>
> **Q1: Will improvements be made on different base model and model sizes?**
>
> Yes, they would! Following your suggestions, we extend our evaluation to include **Qwen-2.5-VL-3B, MIMO-VL-7B, InternVL-2B, InternVL-8B, and TimeSuite-7B**. Due to the time limit, we post-train all base models using our 2.5K filtered by Qwen-2.5-VL-7B. The results are presented as follows:
>
> |||| TVGBench  |||| Charades  |||| ActivityNet |||
> |-|-|-|-|-|-|-|-|-|-|-|-|-|-|
> | Model | Size | R1@0.3| R1@0.5| R1@0.7| mIoU | R1@0.3| R1@0.5| R1@0.7 | mIoU| R1@0.3| R1@0.5 | R1@0.7| mIoU  |
> | Qwen-2.5-VL-3B (Base) | 3B | 11.5 | 6.5  | 3.8  | 8.3  | 24.2  | 15.5 | 8.1| 16.3| 13.0 | 7.1| 3.3| 9.8  |
> | Qwen-2.5-VL-3B (Time-R1) | 3B| 33.5 | 21.0  | 10.5 | 21.7 | 74.6 | 53.1  | 26.0 | 51.2 | 40.0 | 21.0  | 8.7| 23.2|
> | Δ ↑  |  | **+22.0** | **+14.5** | **+6.7** | **+13.4** | **+50.4** | **+37.6** | **+17.9** | **+34.9** | **+27.0** | **+13.9**| **+5.4** | **+13.4** |
> | Qwen-2.5-VL-7B (Base)    | 7B| 24.9  | 16.0  | 8.0  | 16.3  | 58.7 | 38.3 | 16.6 | 37.9      | 34.3 | 21.6  | 12.9  | 22.9 |
> | Qwen-2.5-VL-7B (Time-R1) | 7B   | 41.6| 28.5 | 15.6| 28.6  | 77.4 | 60.0   | 34.1 | 57.2  | 56.2  | 37.4 | 20.4 | 38.0 |
> | Δ ↑ |  | **+16.7** | **+12.5** | **+7.6** | **+12.3** | **+18.7** | **+21.7** | **+17.5** | **+19.3** | **+21.9** | **+15.8**| **+7.5** | **+15.1** |
> | MiMo-VL-7B (Base) | 7B   | 22.4 | 12.6| 6.6  | 15.7  | 48.5 | 27.0 | 12.1  | 31.7 | 31.3  | 19.7  | 12.1  | 24.2 |
> | MiMo-VL-7B (Time-R1)  | 7B   | 41.2   | 27.8 | 15.1| 27.4| 79.9| 63.9| 33.4 | 53.9  | 45.6| 27.2  | 14.2 | 31.9 |
> | Δ ↑  || **+18.8** | **+15.2** | **+8.5** | **+11.7** | **+31.4** | **+36.9** | **+21.3** | **+22.2** | **+14.3** | **+7.5**    | **+2.1** | **+7.7**  |
> | InternVL-2B (Base)  | 2B | 16.3 | 6.3| 2.3 | 11.7  | 20.9   | 7.8  | 1.9  | 15.4 | 18.6  | 8.5   | 3.1 | 14.2 |
> | InternVL-2B (Time-R1)    | 2B   | 21.8 | 9.5  | 4.1 | 14.8 | 24.0 | 11.5 | 3.5  | 15.7 | 20.6  | 9.5  | 3.9  | 14.2 |
> | Δ ↑ | | **+5.5**  | **+3.2**  | **+1.8** | **+3.1**  | **+3.1**  | **+3.7**  | **+1.6**  | **+0.3**  | **+2.0**  | **+1.0**| **+0.8** | **+0.0**  |
> | InternVL-8B (Base) | 8B| 17.4  | 8.3  | 3.4 | 11.8 | 27.8| 11.9 | 3.7 | 20.6  | 33.1| 18.4| 10.3| 24.0 |
> | InternVL-8B (Time-R1)| 8B| 38.0 | 22.5 | 9.2  | 24.2 | 70.0| 45.1 | 18.3| 44.1| 46.8 | 25.9 | 11.7  | 31.1  |
> | Δ ↑  | | **+20.6** | **+14.2** | **+5.8** | **+12.4** | **+42.2** | **+33.2** | **+14.6** | **+23.5** | **+13.7** | **+7.5**| **+1.4** | **+7.1**  |
> | TimeSuite-7B (Base) | 8B| 31.1  | 18.0| 8.8| 21.6 | 70.1 | 48.5 | 24.2 | 45.7 |-|-|-|-|
> | TimeSuite-7B (Time-R1)| 8B| 33.3 | 19.7 | 9.2 | 22.3 | 71.7| 49.5 | 24.2| 47.2 |-|-|-|-|
> | Δ ↑  | | **+2.2** | **+1.7** | **+0.4** | **+0.7** | **+1.6** | **+1.0** | **+0.0** | **+1.5** |-|-|-|-|
>
>
> We observe several key findings, revealing that our approach is generally applied to many base models.
>
> First, **models of different sizes can also benefit from our approach**. For instance, the Qwen-VL-3B model achieves a +13.4 mIoU improvement on TVGBench and a +34.9 mIoU improvement on Charades. This is attributed to the fact that our training data is distilled based on filtering from Qwen-VL-7B, demonstrating that our RL-based method can effectively learn from the data.
>
> Second, **models with different backbones also benefit**. For example, InternVL-8B achieves a +12.4 mIoU improvement on TVGBench and +23.5 on Charades, while MiMo-VL-8B improves by +11.7 and +22.2 on TVGBench and Charades, respectively.
>
> Additionally, we notice that when using a smaller model with a different backbone, the transferability of the training data becomes less effective, but still beneficial. For example, training InternVL-2B on data filtered by Qwen-2.5-VL-7B only yields marginal gains, improving only +3.1 on TVGBench and +0.3 on Charades. Similar phenomenon is observed on the TimeSuite-7B. This is mainly due to architectural differences, and the data filtered by Qwen-2.5-VL-7B are not well-suited for RL training on InternVL-2B. However, we expect that re-filtering the data using InternVL itself could yield improvements comparable to those observed when transferring from Qwen-2.5-VL-7B to Qwen-2.5-VL-3B.
>
> **Q2: Does the improvement in VQA performance come from using VQA data during RL?**
>
> No, and we sincerely apologize for the confusion. As clarified in the Appendix, there was a typo in the main text—we did not use any VQA data during reinforcement learning. The improvement is solely attributed to our proposed method, without reliance on external VQA supervision.
>
> **Q3: Could you provide an ablation study on the reward design?**
>
> Yes, certainly. In response to your request, we provide an comprehensive ablation on the reward design as follows:
> - the CoT template: $r\_{\rm{format}}$
> - original IoU reward: $r\_{\rm{IoU}}=\frac{[t\_s,t\_e]\cap[t\_s',t\_e']}{[t\_s,t\_e]\cup[t\_s',t\_e']}$
> - only the second term of the tIoU-based reward: $r\_{\rm{tIoUsec}}=1-\frac{|t\_s-t\_s'|}{t}\cdot\frac{|t\_e-t\_e'|}{t}$
> - Exact Matching: $r\_{\rm{em}}=1 \rm{if\ t\_s=t\_s'\ and\ t\_e=t\_e'\ else\ 0}$
> - Absolute Error: $r\_{\rm{abs}}=1 - \frac{|t\_s-t\_s'| + |t\_e-t\_e'|}{2t}$
> - RMSE: $r\_{\rm{rmse}}=1 - \sqrt{(\frac{t\_s-t\_s'}{t})^2 + (\frac{t\_e-t\_e'}{t})^2}$
>
> The results, shown below, demonstrate the effectiveness of the tIoU reward design.
>
> | |   | TVGBench |        |      |        | Charades |        |      |        | ActivityNet |        |      |
> |-----------------------------------|--------|----------|--------|------|--------|----------|--------|------|--------|-------------|--------|------|
> | Model                             | R1@0.3 | R1@0.5   | R1@0.7 | mIoU | R1@0.3 | R1@0.5   | R1@0.7 | mIoU | R1@0.3 | R1@0.5      | R1@0.7 | mIoU |
> | $r\_{\rm{tIoU}}+r\_{\rm{format}}$ (ours) | **41.8**   | **29.4**     | **16.4**   | **29.2** | 78.1   | 60.8     | 35.3   | **58.1** | **58.1**  | **39.0**        | **21.4**   | **40.5** |
> | $r\_{\rm{format}}$| 27.1   | 18.0     | 10.1   | 18.8 | 60.4   | 39.2     | 17.4   | 39.3 | 35.1   | 22.2        | 13.1   | 26.0 |
> | $r\_{\rm{tIoU}}$| 40.5   | 27.6     | 15.4   | 28.0 | **78.4**   | **61.6**     | **36.1**   | 54.0 | 55.2   | 36.8        | 20.2   | 38.4 |
> | $r\_{\rm{IoU}}+r\_{\rm{format}}$|41.4 |28| 15.8| 28.4| 77.6   | 58.9| 32.7| 56.4| 58.8| 38.8| 20.7|39.4|
> | $r\_{\rm{em}}+r\_{\rm{format}}$| 26.5   | 16.8     | 9.1    | 18.3 | 61.6   | 40.3     | 18.7   | 40.4 | 35.9   | 22.5        | 13.3   | 26.4 |
> | $r\_{\rm{abs}}+r\_{\rm{format}}$| 39.1   | 27.8     | 14.8   | 27.4 | 76.2   | 59.5     | 33.9   | 52.3 | 51.4   | 32.9        | 17.8   | 35.7 |
> | $r\_{\rm{rmse}}+r\_{\rm{format}}$| 38.9   | 27.0     | 15.8   | 27.2 | 75.4   | 58.6     | 33.1   | 51.5 | 51.2   | 32.7        | 17.8   | 35.5 |
> | $r\_{\rm{tIoUsec}}+r\_{\rm{format}}$| 37.6   | 25.9     | 15.0   | 26.3 | 75.6   | 57.7     | 31.9   | 51.3 | 52.5   | 34.1        | 18.5   | 36.6 |
>
> Experimental results demonstrate that our combination of tIoU and format reward achieves the best overall performance. Without the format reward, the model can still perform reasonably well even without CoT. However, without the tIoU reward, it becomes difficult to effectively activate the model’s TVG capabilities.  While training with $r\_{\rm{tIoU}}$ would lead to better performance on R1@0.3/0.5/0.7 on Charades, the actual mIoU of training with $r\_{\rm{tIoU}}+r\_{\rm{format}}$ outperforms it.

---

> > ### Comment · Reviewer_XTMy · 2025-08-05
> >
> > Thanks for the response, I’m pleased to see that most of my concerns have been addressed. I recommend the authors include the relevant experiments in the revised version. I will maintain my current vote.”

---

> ### Author Response · Authors · 2025-08-07
> **Appreciation to Reviewer XTMy for the Valuable Feedback and Recognition**
>
> Dear Reviewer XTMy,
>
> We sincerely appreciate your recognition of our efforts to address your concerns. We are grateful for your positive feedback and valuable suggestion.
>
> As you recommended, we will incorporate the relevant experiments into the revised version of our paper.
>
> We welcome any further discussion and would be happy to provide additional clarification if needed.
>
> Best regards,
>
> Authors

---

### Note · Authors · 2025-08-14

Dear AC and Reviewers,

Thank you for your valuable time and insightful feedback. We are encouraged by the reviewer's recognition of our clear motivation (**1QoR**), broad contribution (**okJ6**), pioneer paradigm (**iVA4**), reasonable and effective method design (**XTMy, 1QoR**), strong benchmark performance (**XTMy, 1QoR, iVA4**), and well-structured writing (**1QoR, okJ6**).

During the rebuttal phase, we addressed key concerns and further strengthened our work with extensive, well-founded experiments. We sincerely appreciate that all reviewers recognized our efforts and provided positive ratings.

In summary, our rebuttal focused on the following core concerns:
- **Generality of Time-R1 across model sizes and architectures (XTMy, iVA4):** Our additional experiments confirm that the Time-R1 framework is not restricted to the Qwen-2.5-VL-7B model. It consistently and significantly boosts performance across diverse base models and various architectures and sizes, including **Qwen-2.5-VL-3B/7B, InternVL3-2B/8B, MiMo-VL-7B, and TimeSuite-7B**. This demonstrates the robustness and versatility of our RL paradigm as a general post-training framework.
- **Advantage of RL paradigms over SFT (1QoR, iVA4):** We conducted extensive hyperparameter tuning for the SFT setting and scaled training to 339K samples. Even so, SFT underperformed compared to RL trained on only 2.5K samples, highlighting RL’s superior generalization ability and data efficiency.
- **Fine-grained ablations (XTMy, 1QoR, iVA4, okJ6):** We conducted detailed ablation studies on **the tIoU reward design**, **the CoT-format reward**, and **the cold-start mechanism**. The results show that: i) tIoU is the most effective among six candidates. ii) CoT-format reward brings substantial performance gains. iii) Cold-start design greatly improves the performance of smaller models (i.e. the 3B model).

We will integrate all these experiments and analyses into the final version, further clarifying our design rationale, strengthening our technical narrative, and highlighting our contributions.

We believe our work introduces an innovative, effective, and rigorously validated RL paradigm for Temporal Video Grounding (TVG) in long video understanding. We will open-source all code, datasets, and benchmarks to foster progress in the community.

Best regards,

Authors

---

### Decision · Program_Chairs · 2025-09-17

**Decision:**

Accept (poster)

**Comment:**

This paper presents a reinforcement learning (RL) based post-training framework that augments large vision-language models with temporal grounding capability. Initial reviews were mixed, but after the rebuttal all reviewers recommended acceptance. Although RL-based post-training is common in LLMs, the reviewers agreed that its adaptation to vision-language models here is sufficiently innovative. The AC concurs with the reviewers' post-rebuttal assessment and recommend acceptance. The AC encourages the authors to incorporate the rebuttal materials into the camera-ready version.